# Efficient Diffusion Transformer Policies with Mixture of Expert Denoisers for Multitask Learning

**Moritz Reuss**[1,*]**, Jyothish Pari**[2,*]**, Pulkit Agrawal**[2]**, Rudolf Lioutikov**[1]
[1]Intuitive Robots Lab (IRL), Karlsruhe Institute of Technology, Germany
[2]Department of Electrical Engineering and Computer Science (EECS), MIT, USA

## Abstract

Diffusion Policies have become widely used in Imitation Learning, offering several appealing properties, such as generating multimodal and discontinuous behavior. As models are becoming larger to capture more complex capabilities, their computational demands increase, as shown by recent scaling laws. Therefore, continuing with the current architectures will present a computational roadblock. To address this gap, we propose Mixture-of-Denoising Experts (MoDE) as a novel policy for Imitation Learning. MoDE surpasses current state-of-the-art Transformer-based Diffusion Policies while enabling parameter-efficient scaling through sparse experts and noise-conditioned routing, reducing both active parameters by 40% and inference costs by 90% via expert caching. Our architecture combines this efficient scaling with noise-conditioned self-attention mechanism, enabling more effective denoising across different noise levels. MoDE achieves state-of-the-art performance on 134 tasks in four established imitation learning benchmarks (CALVIN and LIBERO). Notably, by pretraining MoDE on diverse robotics data, we achieve 4.01 on CALVIN ABC and 0.95 on LIBERO-90. It surpasses both CNN-based and Transformer Diffusion Policies by an average of $57\%$ across 4 benchmarks, while using 90% fewer FLOPs and fewer active parameters compared to default Diffusion Transformer architectures. Furthermore, we conduct comprehensive ablations on MoDE's components, providing insights for designing efficient and scalable Transformer architectures for Diffusion Policies. Code and demonstrations are available at `https://mbreuss.github.io/MoDE_Diffusion_Policy/`.

## 1 Introduction

Diffusion models learn to reverse an iterative process that adds Gaussian noise to data samples (Ho et al., 2020; Song et al., 2020). After training, they can generate new samples conditioned on goals like instructions or images. Recently, diffusion models have gained widespread adoption as policies for Imitation Learning (IL) (Octo Model Team et al., 2023; Reuss et al., 2023; Chi et al., 2023). IL is a powerful paradigm to train agents from expert demonstrations to learn versatile skills (Pomerleau, 1989; Nair et al., 2017; Pari et al., 2021; Fu et al., 2024).

Diffusion policies offer several appealing properties for IL: they can generate diverse multimodal behavior (Jia et al., 2024), scale with more data (Octo Model Team et al., 2023), and handle discontinuities in the action space (Chi et al., 2023). However, a major limitation is their high computational cost, resulting in slower training and inference speed as models become larger. Standard architectures contain hundreds of millions of parameters (Chi et al., 2023) and require many denoising steps to generate actions. Large encoder modules for images and text further increase the computational requirements for IL policies. This restricts their use in real-time robotics applications, particularly in scenarios with limited on-board computing resources, such as mobile robots.

To address these challenges, we explore Mixture-of-Experts (MoE) that can scale model capacity while using fewer FLOPs for training and inference. The core idea behind a sparse MoE is to utilize only a subset of the total model parameters during each forward pass. This is achieved by having

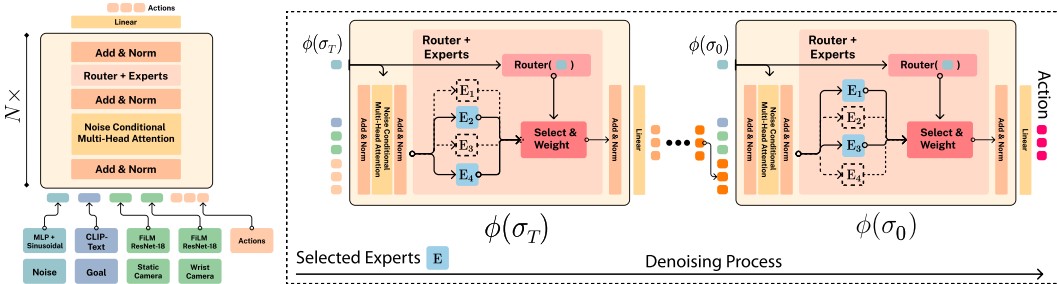

Figure 1: The proposed MoDE architecture (left) uses a transformer with causal masking, where each block includes noise-conditional self-attention and a noise-conditioned router that assigns tokens to expert models based on the noise level. This design enables efficient, scalable action generation. On the right, the router's activation of subsets of simple MLP experts with Swish-GLU activation during denoising is illustrated.

multiple expert subnetworks and a routing model, that sparsely activates experts and interpolates their outputs, based on the input.

We introduce **M**ixture-of-**D**enoising **E**xperts Policy (MoDE), a scalable and efficient MoE Diffusion Policy. Our work is inspired by prior results showcasing the multitask nature of the denoising process (Hang et al., 2024), where there is little transfer between the different phases in the denoising process. We present a novel noise-conditioned routing mechanism, that distributes tokens to our experts based on the current noise level. MoDE leverages noise-conditioned self-attention combined with a noise input token for enhanced noise-injection. Our proposed Policy surpasses previous Diffusion Policies with higher efficiency and demonstrates sota performance across 134 diverse tasks in challenging goal-conditioned imitation learning benchmarks: CALVIN (Mees et al., 2022b) and LIBERO (Liu et al., 2023). Through comprehensive ablation studies, we investigate the impact of various design decisions, including token routing strategies, noise-injection techniques, expert distribution and diverse pretraining on a large-scale robot dataset (Collaboration et al., 2023). We summarize our contributions below:

- We introduce MoDE, a novel Mixture-of-Experts Diffusion Policy that achieves state-of-the-art performance while using $90\%$ fewer FLOPs and less active parameters than dense transformer baselines thanks to our noise-based expert caching and sparse MoE design.

- We demonstrate MoDE's effectiveness across 134 tasks in 4 benchmarks, showing an average $57\%$ performance increase over prior Diffusion Policies while maintaining improved computational efficiency.

- We present detailed ablation studies that investigate the importance of routing strategies and noise-injection, visualizing expert utilization across denoising steps to identify key components of MoDE.

## 2 RELATED WORK

**Diffusion in Robotics.** In recent years, Diffusion Models (Song & Ermon, 2019; Ho et al., 2020; Karras et al., 2022) have gained widespread adoption in the context of robotics. They are used as a policy representation for Imitation Learning (Chi et al., 2023; Reuss et al., 2024; Xian et al., 2023; Ke et al., 2024; Li et al., 2023b; Scheikl et al., 2023) and in Offline Reinforcement Learning (Ajay et al., 2023a; Janner et al., 2022; Pari et al., 2022). Other applications of diffusion models in robotics include robot design generation (Wang et al., 2023), video-generation (Du et al., 2023; Ko et al., 2023; Ajay et al., 2023b) and motion planning (Carvalho et al., 2023; Urain et al., 2023). The most common architecture for using diffusion models as a policy in robotics is a Convolutional neural network (CNN) with additional FiLM conditioning (Perez et al., 2018) to guide the generation based on context information. Recently, the transformer architecture has been adopted as a strong alternative

---

[1]Correspondence to: moritz.reuss@kit.edu

[2]Both authors contributed equally.

backbone for diffusion policies, specifically in IL. Examples include Octo (Octo Model Team et al., 2023), BESO (Reuss et al., 2023) and 3D-Diffusion-Actor (Ke et al., 2024). Sparse-DP leverages MoE to learn specialist experts for different tasks (Wang et al., 2024). However, no prior work considers using a Mixture of Experts architecture for improving the computational efficiency and inference time of the denoising process for generalist policies.

**Mixture-of-Experts.** MoE are a class of models where information is selectively routed through the model. The modern version of MoE was introduced in (Shazeer et al., 2017), where a routing or gating network conditionally chooses a subset of experts to send an input to. After Transformers (Vaswani et al., 2017) proved to be an effective model that scales well with data, they were modified to have expert feed-forward networks at each block of the model in (Fedus et al., 2022) which presented Switch Transformers. Switch Transformers laid the groundwork that is still widely adopted in different Large-Language-Models (LLM) (Jiang et al., 2024; Du et al., 2022). This allowed for more total parameters while keeping the forward and backward FLOPs smaller than their dense counterpart, thus yielding significant performance gains. However, training both the router and experts in parallel is a non-trivial optimization problem, and it can yield suboptimal solutions such as expert collapse where experts learn similar functions instead of specializing (Chi et al., 2022). In addition, router collapse occurs when the router selects a small subset of the experts and doesn't utilize all the experts. This is mitigated with load balancing losses (Shazeer et al., 2017; Fedus et al., 2022) which encourage the router to distribute inputs more evenly across experts. Multiple works have explored different methods to perform routing, such as expert choice routing (Zhou et al., 2022), differential k-selection (Hazimeh et al., 2021), frozen hashing functions (Roller et al., 2021), and linear assignment (Lewis et al., 2021).

**Multi Task Learning in Diffusion Models.** It has been shown that the denoising process is multi-task Hang et al. (2024). Leveraging this idea, works have adopted architectures suited for multi-task learning. Some works have explicitly scheduled which parameters are specialized to which stage in the denoising process (Park et al., 2023; Go et al., 2023). In extension to this (Park et al., 2024) uses the scheduling as guidance during training but also learns how to modulate representations based on the denoising stage. Finally, some works have employed different architectures based on the denoising stage (Lee et al., 2024b).

## 3 METHOD

In this section, we introduce MoDE, our novel MoE Diffusion Policy. First, we formulate the problem of learning a policy for IL. Next, we summarize the framework used in Diffusion Policies and then introduce our MoDE architecture with our novel noise-conditioned routing and noise-conditioned self-attention that enable efficient policy design. Finally, we explain our expert caching mechanism for efficient inference and explain the pretraining of MoDE.

### 3.1 PROBLEM FORMULATION

We consider the problem of learning a language-conditioned policy $\pi_\theta(\bar{a}|\bar{s}, g)$ given a dataset of robot demonstrations $\mathcal{T}$. The policy predicts a sequence of future actions $\bar{a} = (a, \ldots, a_{i+j-1})$ of length $j$, conditioned on the history of state embeddings $\bar{s} = (s_{i-h+1}, \ldots, s_i)$ of length $h$ and a desired goal $g$. The dataset contains $\tau \in \mathcal{T}$ trajectories, where trajectory consists of a sequence of triplets of state, actions, and goal $(\bar{s}_i, a_i, g)$. $g$ is a language instruction. Our policy is trained to maximize the log-likelihood of the action sequence given the context of state history and goal:

$$\mathcal{L}_{\text{IL}} = \mathbb{E}\left[ \sum_{(\bar{s}, \bar{a}, g) \in \mathcal{T}} \log \pi_\theta\left(\bar{a}|\bar{s}, g\right) \right]. \tag{1}$$

### 3.2 DIFFUSION POLICY

MoDE uses the continuous-time diffusion model of EDM (Karras et al., 2022) as a policy representation. Diffusion models are a type of generative model for generating data by initially adding noise through Gaussian perturbations and then reversing this process. MoDE applies the score-based diffusion model to represent the policy $\pi_\theta(\bar{a}|\bar{s}, g)$. The perturbation and inverse process can be

described using a stochastic differential equation:

$$d\bar{a} = (\beta_t \sigma_t - \dot{\sigma}_t)\sigma_t \nabla_a \log p_t(\bar{a}|\bar{s}, g)dt + \sqrt{2\beta_t}\sigma_t d\omega_t, \tag{2}$$

where $\beta_t$ controls the noise injection, $d\omega_t$ refers to infinitesimal Gaussian noise, and $p_t(\bar{a}|\bar{s}, g)$ is the score function of the diffusion process, that moves samples away from regions of high-data density in the forward process. To generate new samples from noise a neural network is trained to approximate the score function $\nabla_{\bar{a}} \log p_t(\bar{a}|\bar{s}, g)$ via Score matching (SM) (Vincent, 2011)

$$\mathcal{L}_{\text{SM}} = \mathbb{E}_{\sigma, \bar{a}, \epsilon}\big[\alpha(\sigma_t)\|D_\theta(\bar{a} + \epsilon, \bar{s}, g, \sigma_t) - \bar{a}\|_2^2\big], \tag{3}$$

where $D_\theta(\bar{a} + \epsilon, \bar{s}, g, \sigma_t)$ is the trainable neural network. During training, we sample noise from a training distribution and add it to an action sequence. The network predicts the denoised actions and computes the SM loss.

After training, we can generate a new action sequence starting from random noise by approximating the reverse SDE or related ODE in discrete steps using numerical ODE integrators. Therefore, we sample noise from the prior $a_T \sim \mathcal{N}(0, \sigma_T^2 \mathbf{I})$ and iteratively denoise it. MoDE uses the DDIM-solver, a numerical ODE-solver designed for diffusion models (Song et al., 2021), that allows fast denoising of actions in a few steps. MoDE uses 10 denoising steps in all our experiments.

## 3.3 MIXTURE-OF-EXPERTS DENOISING

We now introduce MoDE, a novel approach that employs noise-conditioned expert routing to enhance diffusion-based policies. This novel routing mechanism enables us to precompute and fuse the required experts for more efficient inference. An overview of MoDE's architecture with the routing mechanism is shown in Figure 1.

For language conditioning, MoDE leverages a frozen CLIP language encoder to generate a latent goal vector, while image encoding utilizes FiLM-conditioned ResNets-18/50. The model processes a sequence of input tokens $\mathbf{X} \in \mathbb{R}^{\text{tokens} \times \mathbf{D}}$ and noise level $\sigma_t$. A linear projection layer $\phi(\sigma_t)$ encodes the noise level into a token, which is incorporated into $\mathbf{X}$. The complete MoDE architecture, $\text{MoDE}(\mathbf{X}, \phi(\sigma_t))$, consists of $L$ transformer blocks, each specialized for different denoising phases.

We now define each block $f^i$ as a composition of a self-attention (SA) layer and an MoE layer,

$$f^i(\mathbf{X}, \phi(\sigma_t)) = \text{MoE}(\text{SA}(\hat{\mathbf{X}}) + \mathbf{X}, \phi(\sigma_t)) + \mathbf{X}. \tag{4}$$

A key change in our approach is the integration of noise-aware positional embeddings. By adding $\phi(\sigma_t)$ to all tokens in $\mathbf{X}$ before self-attention:

$$\hat{\mathbf{X}} = \phi(\sigma_t) + \mathbf{X}, \tag{5}$$

we enable each token to adapt its attention patterns based on the current denoising phase. This design enhances denoising performance without introducing additional parameters or architectural complexity.

The self-attention mechanism follows the standard formulation (Vaswani et al., 2017):

$$\text{SA}(\hat{\mathbf{X}}) = \text{softmax}(\frac{1}{\sqrt{D}}[\hat{\mathbf{X}}W_Q][\hat{\mathbf{X}}W_K]^T)[\hat{\mathbf{X}}W_V]. \tag{6}$$

Our MoE layer introduces a novel noise-conditioned routing strategy. Given $N$ experts $\mathbf{E_i}i = 1^N$, the layer output is:

$$\text{MoE}(\mathbf{X}, \phi(\sigma_t)) = \sum_{i=1}^{N} \mathbf{R}(\phi(\sigma_t))\mathbf{E_i}(\mathbf{X}), \tag{7}$$

where the routing function $\mathbf{R}(\cdot) : \mathbb{R}^{\text{tokens} \times \mathbf{D}} \to \mathbb{R}^{\text{tokens} \times \mathbf{N}}$ determines expert selection:

$$\mathbf{R}(\phi(\sigma_t)) = \text{topk}(\text{softmax}(\phi(\sigma_t)\mathbf{W_R}), k) \tag{8}$$

Unlike traditional MoE approaches that route based on input content, MoDE's routing mechanism specifically considers the noise level. This enables specialized experts for different denoising phases,

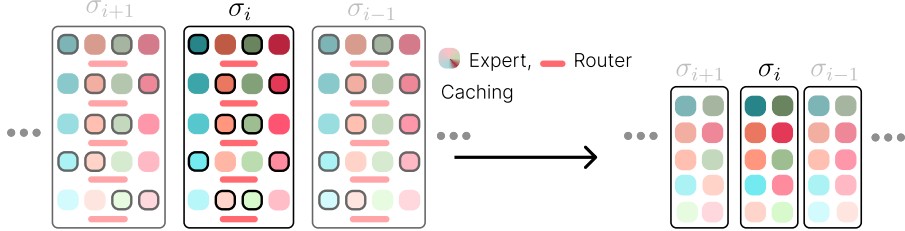

Figure 2: After training MoDE, the router is noise-conditioned, allowing pre-computation of the experts used at each noise level before inference. This enables removing the router and retaining only the selected experts, significantly improving network efficiency.

allowing for both improved performance and computational efficiency through expert caching (detailed in subsubsection 3.3.1). We utilize the same method from (Muennighoff et al., 2024) to initialize the router, which is from a truncated normal distribution with std $= 0.02$. We implement topk using multinomial sampling without replacement, selecting $k$ elements according to softmax$(\phi(\sigma_t)W_R)$ probabilities. To maintain gradient flow through the non-differentiable sampling process, we scale expert outputs by routing probabilities and renormalize the selected probabilities. To prevent expert collapse, we incorporate a load balancing loss ($LB$) (Fedus et al., 2022):

$$LB(\sigma_t) = N \sum_{n=1}^{N} \frac{1}{|\mathcal{B}|} (\sum_{i=1}^{|\mathcal{B}|} \mathbb{1}\mathbf{R}(\phi(\sigma_{\mathbf{t_i}}))\mathbf{n} > \mathbf{0}) \frac{1}{|\mathcal{B}|} (\sum i = 1^{|\mathcal{B}|} \text{softmax}(\phi(\sigma_{t_i})W_R)_n) \quad (9)$$

with $\gamma = 0.01$ as the balancing coefficient.

### 3.3.1 ROUTER AND EXPERT CACHING

To make our method more efficient, we exploit the fact that our MoE is noise-conditioned, meaning where at each noise level, the routing path is deterministic and can be precomputed. By doing so, we can determine the chosen experts ahead of time for each noise level. Figure 2 illustrates this process. This allows us to fuse the selected expert MLPs into a single, composite MLP, effectively reducing the computation cost. Instead of looping through each expert individually, this fused expert MLP can run in parallel, which substantially decreases the overall latency of the network. By eliminating the need for dynamically invoking each expert, we not only save time but also streamline memory access patterns, reducing the overhead typically associated with routing decisions in a traditional MoE setup. Our caching strategy reduces the FLOPs overhead by over 80% compared to standard MoE rollouts and makes it two times faster during inference.

### 3.4 GENERALIST POLICY PRE-TRAINING

We pretrain MoDE on a diverse mix of multi-robot datasets curated from the OXE dataset Collaboration et al. (2023). Our training data encompasses 196k trajectories selected from six diverse datasets, featuring various robot platforms and manipulation tasks. The pretraining process of MoDE runs for 300k steps on a distributed Cluster of 6 GPUs over three days For finetuning we freeze the pretrained routers in each layer and only finetune the other model components. A comprehensive overview of our pretraining dataset composition and methodology is provided in the Appendix (subsection A.1).

In detailed evaluations using the real2sim benchmark SIMPLER(Li et al., 2024b), MoDE demonstrates superior performance compared to state-of-the-art generalist policies. It achieves an average success rate of 26.30% across diverse manipulation tasks, outperforming both OpenVLA Kim et al. (2024) (23.70%) and Octo Octo Model Team et al. (2023) (17.75%). Full evaluation details are provided in subsubsection A.2.1.

## 4 EVALUATION

Our experiments aim to answer four key questions: (I) How does MoDE compare to other policies and prior diffusion transformer architectures in terms of performance? (II) Does large-scale pre-training

of diverse robotics data boost the performance of MoDE? (III) What is MoDE's efficiency and speed compared to dense transformer baselines? (IV) How does the model utilize different experts during the action-denoising process?

We compare MoDE against prior diffusion transformer architectures (Chi et al., 2023), ensuring fair comparisons by using a similar number of active parameters. MoDE uses 8 layers with 4 experts and a latent dimension of 1024 in all experiments. Our pretrained variant is slightly bigger with 12 layers and 4 experts with the same latent dimension of 1024.

We use an action chunking length of 10 and a history length of 1 for all experiments. MoDE does execute all 10 generated actions without early re-planning or temporal aggregation. Detailed hyperparameters are provided in the Appendix ( Table 3).

### 4.1 LONG-HORIZON MULTI-TASK EXPERIMENTS

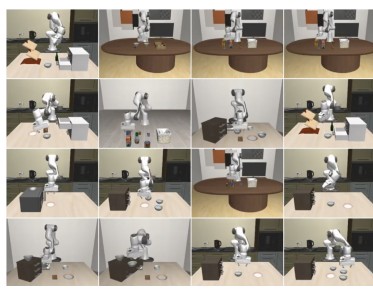

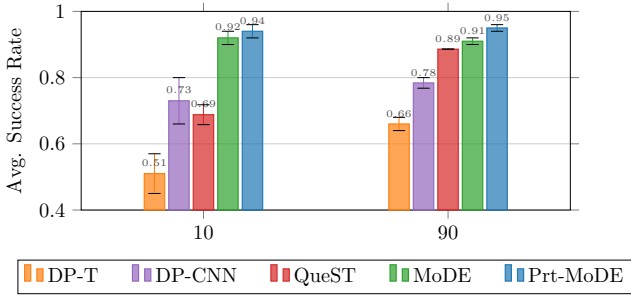

(a) LIBERO-90 Tasks                    (b) Results for LIBERO-10 and LIBERO-90

Figure 3: Visualization and Results for LIBERO environment. (a) Few example environments and tasks of the LIBERO-90 task suite. (b) Average results for both LIBERO challenges averaged over 3 seeds with 20 rollouts for each task.

We first evaluate MoDE on the LONG-challenge and LIBERO-90 challenge of the LIBERO benchmark (Liu et al., 2023). The LONG challenge requires a model to learn 10 tasks in different settings. It demands long-horizon behavior generation with several hundreds of steps for completion. The 90 variant tests policies on 90 diverse short-horizon tasks in different environments. Figure 3a visualizes a few examples of these tasks. All environments feature two cameras: a static one and a wrist-mounted camera, used to encode the current observation using FiLM-ResNets-18. We test each policy 20 times on each task and report the average results over 3 seeds. MoDE and all other diffusion architectures use FiLM-conditioned ResNets-18 with a CLIP sentence embedding to encode the goal and the images.

**Baselines.** We compare MoDE against three state-of-the-art baselines: 1) The Diffusion Transformer (DP-T) architecture (Chi et al., 2023), which conditions on noise and observations using a cross-attention module. 2) The standard Diffusion Policy CNN-based architecture (DP-CNN). 3) QueST (Mete et al., 2024), a transformer-based policy that learns discrete action representations using vector-quantized embeddings of action sequences. We tested all baselines ourselves, except for QueST, whose results are taken directly from their paper.

**Results.** The performance of all models on the benchmark is summarized in Figure 3b. Overall, MoDE achieves the highest average performance in both benchmarks, while the QueST baseline is the second best in the LIBERO-90 setting and the CNN-architecture is second best in the long horizon setting. These results demonstrate MoDE's ability to learn long-horizon tasks with high accuracy. The performance gap is more pronounced in the challenging LIBERO-10 experiment, where MoDE is the first policy to achieve an over 90% success rate. Furthermore, MoDE surpasses prior best Diffusion baselines by an average of 16% in both settings, all while maintaining its computational advantage. The pretrained MoDE variant achieves an even higher performance in both settings, showing the potential of diverse pretraining. This showcases MoDE's ability to achieve state-of-the-art performance with a more efficient use of computational resources.

| Train→Test | Method | Active Params in Million | PrT | No. Instructions in a Row (1000 chains) | | | | | |
|---|---|---|---|---|---|---|---|---|---|
| | | | | 1 | 2 | 3 | 4 | 5 | Avg. Len. |
| ABCD→D | DP-CNN | 321 | × | 86.3% | 72.7% | 60.1% | 51.2% | 41.7% | 3.16±0.06 |
| | DP-T | 194 | × | 78.3% | 53.9% | 33.8% | 20.4% | 11.3% | 1.98±0.09 |
| | RoboFlamingo | 1000 | ✓ | 96.4% | 89.6% | 82.4% | 74.0% | 66.0% | 4.09±0.00 |
| | GR-1 | 130 | ✓ | 94.9% | 89.6% | 84.4% | 78.9% | 73.1% | 4.21±0.00 |
| | **MoDE** | 277 | × | 96.6% | 90.6% | 86.6% | 80.9% | 75.5% | 4.30±0.02 |
| | **MoDE** | 436 | ✓ | **97.1%** | **92.5%** | **87.9%** | **83.5%** | **77.9%** | **4.39±0.04** |
| ABC→D | DP-CNN | 321 | × | 63.5% | 35.3% | 19.4% | 10.7% | 6.4% | 1.35±0.05 |
| | DP-T | 194 | × | 62.2% | 30.9% | 13.2% | 5.0% | 1.6% | 1.13±0.02 |
| | RoboFlamingo | 1000 | ✓ | 82.4% | 61.9% | 46.6% | 33.1% | 23.5% | 2.47±0.00 |
| | SuSIE | 860+ | ✓ | 87.0% | 69.0% | 49.0% | 38.0% | 26.0% | 2.69±0.00 |
| | GR-1 | 130 | ✓ | 85.4% | 71.2% | 59.6% | 49.7% | 40.1% | 3.06±0.00 |
| | **MoDE** | 307 | × | 91.5% | 79.2% | 67.3% | 55.8% | 45.3% | 3.39±0.03 |
| | **MoDE** | 436 | ✓ | **96.2%** | **88.9%** | **81.1%** | **71.8%** | **63.5%** | **4.01±0.04** |

Table 1: Performance comparison on the two CALVIN challenges. The table reports average success rates for individual tasks within instruction chains and the average rollout length (Avg. Len.) to complete 5 consecutive instructions, based on 1000 chains. Zero standard deviation indicates methods without reported average performance. "Prt" denotes models requiring policy pretraining. Parameter counts exclude language encoders.

## 4.2 SCALING MULTI-TASK EXPERIMENTS

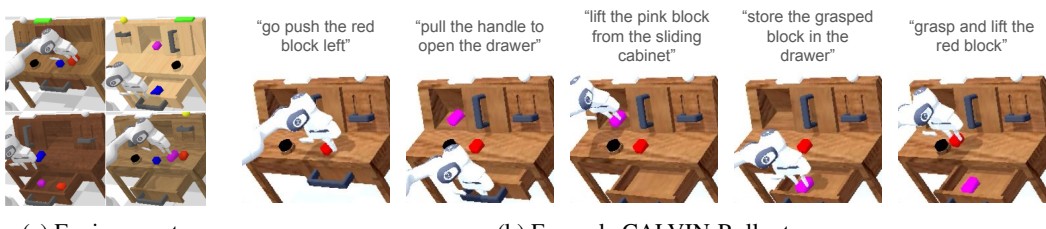

(a) Environments             (b) Example CALVIN-Rollout

Figure 4: Overview of the CALVIN environment. (a) CALVIN contains four different settings (A,B,C,D) with different configurations of slides, drawers and textures. (b) Example rollout consisting of 5 tasks in sequence. The next goal is only given to the policy, if it manages to complete the prior.

We evaluate MoDE on the **CALVIN Language-Skills Benchmark** (Mees et al., 2022b), a comprehensive image-based benchmark for IL. In the **ABCD→D** challenge, models learn from 22, 966 interaction sequences across four environments (A, B, C, D), each comprising 64 timesteps and 34 diverse tasks requiring complex skill chaining (Figure 4a). Following the official protocol, we test all policies on 1000 instruction chains of 5 sequential tasks in environment D (Figure 4b). Models receive 1 point per completed task and must complete each task to progress. We report average sequence length over 3 seeds with 1000 instruction chains each.

**Baselines.** We test MODE against several methods specialized for learning language-conditioned behavior and against other baseline diffusion policy architectures. We also compare MoDE against RoboFlamingo and GR-1. RoboFlamingo is a fine-tuned Vision-Language-Action model, that contains around 3 billion parameters and has been pre-trained on diverse internet data. GR-1 is a generative decoder-only Transformer pretrained on large-scale video generation and then co-finetuned on CALVIN (Wu et al., 2024). If available, we report the average performance of all prior work directly from their paper, given the standard evaluation protocol in CALVIN (Mees et al., 2022a).

**Results.** Our findings (Table 1) show that MoDE achieves the highest average success rate across all policies, even without pretraining. This surpasses extensively pretrained baselines like RoboFlamingo and GR-1. While GR-1 uses fewer parameters (130M vs 436M), MoDE requires significantly fewer FLOPs during inference (1.53 vs 27.5 GFLOPS) while maintaining comparable speed (12.2 vs 12.6 ms). The combination of state-of-the-art performance and computational efficiency makes MoDE particularly suitable for multitask applications.

### 4.3 ZERO-SHOT GENERALIZATION EXPERIMENTS

Finally, we then extend our investigation to the **ABC→D** challenge in the second phase, testing MoDE's zero-shot generalization abilities. In this experiment, models are only trained on data from the first three CALVIN environments A,B,C and tested on the unseen setting of environment D, which has different positions of relevant objects and texture of the table. This requires policies, that are able to generalize their learned behavior to new environment configurations and different textures, which is especially challenging. We evaluate MoDE trained from scratch and MoDE pretrained on a sub-set of Open-X-Embodiment Data. This enables us to study the zero-shot performance and pretraining effectiveness of MoDE.

**Baselines.** For this experiment, we compare MODE against the previous CALVIN baselines, with the addition of SuSIE (Black et al., 2023). A hierarchical policy utilizing a finetuned image-generation model, Instruct2Pix (Brooks et al., 2023), to generate goal images, which guide a low-level diffusion policy. The high-level goal generation model requires large-scale pretraining. SuSIE's results are based on 100 rollouts only, without standard deviation, due to the computational cost of generating subgoal images.

**Results.** The results of this experiment are summarized in the lower part of Table 1. MoDE outperforms all tested baselines and surpasses all other Diffusion Policy architectures by a wide margin. Further, by pretraining MoDE on diverse robotics data, we achieve a new sota performance of 4.01. Therefore, in response to Question (I), we affirmatively conclude that Mixture-of-Experts models not only enhance scaling performance but also demonstrate strong zero-shot generalization capabilities. In addition, we can answer Question (II) by concluding that pretraining boost performance in challenging zero-shot settings.

### 4.4 COMPUTATIONAL EFFICIENCY OF MoDE

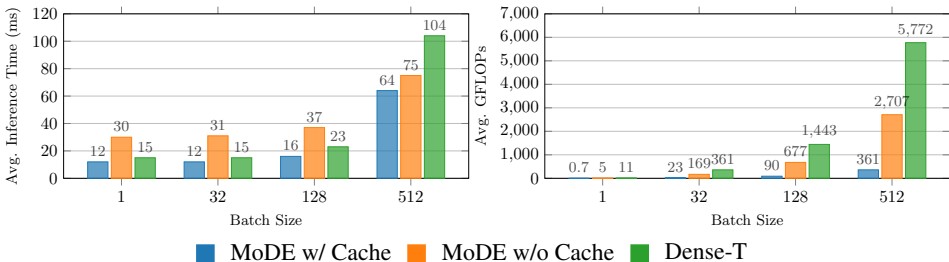

Figure 5: Computational efficiency comparison between MoDE and a Dense-Transformer model with the same number of parameters. Left: Average inference speed over 100 forward passes for various batch sizes. Right: FLOP count for MoDE with router cache and without compared against a dense baseline. MoDE demonstrates superior efficiency with lower FLOP count and faster inference thanks to its router caching and sparse expert design.

We compare MoDE against a dense transformer baseline with similar parameters by measuring average inference time and FLOPs across batch sizes. As shown in Figure Figure 5, MoDE with caching significantly improves computational efficiency - at batch size 1, inference is 20% faster (12ms vs 15ms), while at batch size 512, MoDE requires 16x fewer FLOPs (361 vs 5,772 GFLOPS) and achieves nearly 40% faster inference (64ms vs 104ms). These results demonstrate that MoDE delivers both superior task performance and substantial computational efficiency through its architecture and caching mechanism. A detailed comparison of inference speed and FLOPS against all other baselines on CALVIN is summarized in subsection A.5.

### 4.5 ABLATION STUDIES

To thoroughly evaluate MoDE's design choices, we conducted a series of ablation studies. These experiments address our research questions: the computational efficiency of MoDE (Question III), the impact of routing strategies (Question IV), and the token distribution (Question V).

### 4.5.1 WHAT DESIGN DECISIONS AFFECT MODE'S PERFORMANCE?

First, we assess the impact of various design decisions on MoDE's performance. We ablate the choice of noise-conditioning and various MoE strategies on the LIBERO-10 benchmark. The results are summarized in Table 2.

**Noise-Injection Ablations.** Our experiments reveal the importance of proper noise conditioning in MoDE. The full MoDE model, which uses both input noise tokens and noise-conditioned self-attention, achieves the best performance with an average success rate of $0.92$. Removing the input noise token slightly decreases performance to $0.90$, highlighting the complementary nature of both conditioning methods. Using only the noise token for conditioning, without noise-conditioned self-attention, further reduces performance to $0.85$. Interestingly, using FiLM conditioning (Perez et al., 2018), a common approach in image-diffusion (Peebles & Xie, 2023), yields the lowest performance in this group at $0.81$. These results underscore the effectiveness of our proposed noise conditioning strategy in MoDE, demonstrating a clear performance advantage of $0.08$ over the FiLM approach.

**MoE Ablations.** Next, we ablate several design decisions regarding Mixture-of-Experts. First, we test the topk number of used experts. Setting topk to 1 only marginally lowers the average performance from $0.92$ to $0.91$. MoDE maintains robust performance even with a single expert. We also examine the effect of using a shared expert, where the model consistently employs the same expert in all cases. This approach achieves a comparable average success rate of $0.90$. Different choices for the token-distribution loss are also tested. While MoDE uses $\gamma = 0.01$ as a default value, we experiment with $\gamma$ values of $0.1$ and $0.001$, which result in average success rates of $0.90$ and $0.86$, respectively. These results indicate that a $\gamma$ value of $0.01$ strikes the best performance.

|  | Avg. Success. |
|---|---|
| **MoDE** | **0.92** |
| - Input Noise Token | 0.90 |
| - Noise-cond Attention | 0.85 |
| FiLM Noise Conditioning | 0.81 |
| TopK=1 | 0.91 |
| Shared Expert | 0.90 |
| $\gamma = 0.1$ | 0.90 |
| $\gamma = 0.001$ | 0.86 |
| 256 Embed Dim | 0.86 |
| 512 Embed Dim | 0.87 |

Table 2: Ablation Studies for MoDE on LIBERO-10. All results are averaged over 3 seeds with 20 rollouts each.

**Latent Dimension.** We investigate the impact of varying the latent dimension in MoDE, testing dimensions of 256, 512, and 1024 (our default). The results show that increasing the latent dimension from 256 to 512 yields a modest performance improvement from $0.86$ to $0.87$, while further increasing to 1024 provides a more substantial boost to $0.92$. This suggests that a larger latent dimension allows MoDE to capture more complex representations, leading to improved performance.

### 4.5.2 OPTIMAL ROUTING STRATEGY FOR DIFFUSION TRANSFORMERS

Next, we answer Question (III) by testing different routing strategies for our diffusion-transformer policy across several environments. We test two different token routing strategies: 1) Token-only conditioned Routing and 2) Noise-only Token Routing. (1) is commonly used in LLMs, where the routing is decided based on the tokens only. We test these strategies in five experiments and report the average performance over 3 seeds: Noise-only Routing achieves an average normalized performance of $0.851$, slightly outperforming Token-only Routing, which achieves $0.845$. Detailed results are summarized in Table 7 in the Appendix. The results demonstrate the effectiveness of our proposed routing strategy. While the performance difference is small, Noise-only Routing offers an additional advantage: the ability to predict all used experts based on noise levels once before roll-outs, enabling faster inference, as described in subsubsection 3.3.1. This is particularly beneficial for robotics applications.

### 4.5.3 HOW DOES THE MODEL DISTRIBUTE THE TOKENS TO DIFFERENT EXPERTS?

To address Question IV, we analyzed how MoDE distributes tokens to different experts using a pre-trained model. Figure 6 visualizes the average usage of each expert in each model layer during inference across various noise levels, using 10 denoising steps for clarity. Our analysis reveals that MoDE learns to utilize different experts for various noise levels, suggesting that the router has specialized for different noise regimes. A transition in expert utilization occurs around $\sigma_8$. In the first layer, the model learns an expert specialized for low-noise levels, primarily used in the last denoising step at $\sigma_{\min}$. We conduct more ablations studies with our pretrained model and various other versions

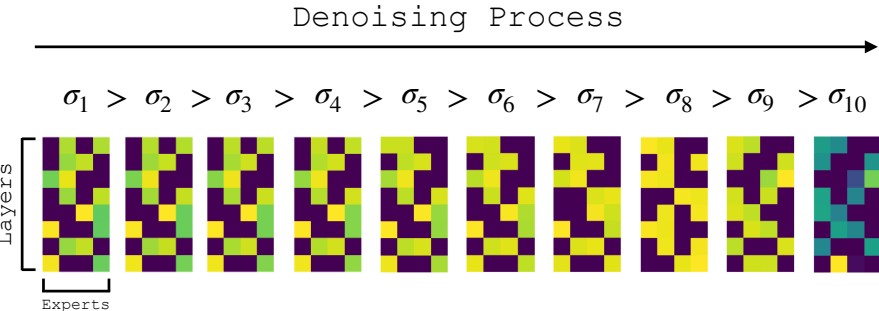

Figure 6: Visualized Expert Utilization. The average usage of all experts in MoDE across all layers is shown. Purple color corresponds to low usage and yellow color to high one, and each image is separately normalized. The average activation shows that MoDE learns to utilize different experts for different noise levels.

of MoDE in subsubsection A.7.1 of the Appendix. These findings affirmatively answer Question IV, demonstrating that MoDE effectively distributes tokens across experts based on noise levels.

### 4.5.4 HOW DOES THE MODEL SCALE WITH MORE EXPERTS?

Finally, we analyze the effect of increasing the number of experts in MoDE. The results are presented in Figure 8 in the Appendix, where we evaluate MoDE on the CALVIN ABCD and CALVIN ABC benchmarks using 2, 4, 6, and 8 experts. For comparison, we include two dense MoDE baselines: Dense-small and Dense-large. Dense-small shares the same latent dimensionality as MoDE, while Dense-large is scaled up to 2024 dimensions, matching MoDE's overall parameter count. Our analysis focuses on how scaling affects both general performance (C-ABCD) and zero-shot generalization (C-ABC). In the ABCD environment, MoDE with 4 experts achieves the best performance. Interestingly, increasing beyond 4 experts degrades performance, possibly due to overfitting or increased routing complexity. In the zero-shot generalization (ABC), MoDE with 4 experts still performs best. Notably, the Dense-small variant consistently underperforms across both tasks, underscoring the efficiency of the MoE architecture in utilizing parameters more effectively. We hypothesize, that 4 experts has an ideal trade-off for the context of noise-only routing of Diffusion Policies. The different expert specialization patterns observed in Figure 6 and Figure 12 from the Appendix show that expert specialization are based on noise regions. MoDE with more than 4 experts does not have a performance benefit. Overall, MoDE demonstrates that it can achieve comparable or superior performance to dense transformer models while requiring fewer computational resources.

### 4.6 LIMITATIONS

MoDE still has certain limitations. In our experiments, we find that MoDE exhibits a slightly higher standard deviation compared to the baselines. We hypothesize that the router's initialization has significant impact on overall optimization, requiring future work on stabilizing routing models. In addition, when visualizing expert utilization, in some of our experiments we noticed that only a subset of the total experts were being utilized - a phenomenon known as expert collapse (Chi et al., 2022).

## 5 CONCLUSION

In this work, we introduced Mixture-of-Denoising Experts (MoDE), a novel Diffusion Policy that leverages a mixture of experts Transformer to enhance the performance and efficiency of diffusion policies. We also proposed a noise-conditioned routing strategy for learning specialized experts within our model. In our extensive experiments and ablation studies across diverse benchmarks, we demonstrated the advantages of MoDE to outperform prior Diffusion Policies with a lower number of parameters and $90\%$ less FLOPS during inference. In future work, we want to experiment with more routing strategies, such as expert-choice routing (Zhou et al., 2022).

## 6 ACKNOWLEDGMENTS

We would like to thank Adam Wei, Anurag Ajay, Hao-Shu Fang, Anthony Simeonov, Yilun Du for their insightful discussions and feedback. The work was funded by the German Research Foundation (DFG) – 448648559. The authors also acknowledge support by the state of Baden-Württemberg through HoreKa supercomputer funded by the Ministry of Science, Research and the Arts Baden-Württemberg and by the German Federal Ministry of Education and Research. The research was also sponsored by the Army Research Office and was accomplished under ARO MURI Grant Number W911NF-23-1-0277.

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

## A    APPENDIX / SUPPLEMENTAL MATERIAL

| Hyperparameter | CALVIN ABCD | CALVIN ABC | LIBERO-10 | LIBERO-90 | Pret-MoDE |
|---|---|---|---|---|---|
| Number of Transformer Layers | 8 | 8 | 8 | 8 | 12 |
| Number Experts | 4 | 4 | 4 | 4 | 4 |
| Attention Heads | 8 | 8 | 8 | 8 | 8 |
| Action Chunk Size | 10 | 10 | 10 | 10 | 10 |
| History Length | 1 | 1 | 1 | 1 | 1 |
| Embedding Dimension | 1024 | 1024 | 1024 | 1024 | 1024 |
| Image Encoder | FiLM-ResNet18 | FiLM-ResNet50 | FiLM-ResNet18 | FiLM-ResNet18 | FiLM-ResNet50 |
| Goal Lang Encoder | CLIP ViT-B/32 | CLIP ViT-B/32 | CLIP ViT-B/32 | CLIP ViT-B/32 | CLIP ViT-B/32 |
| Attention Dropout | 0.3 | 0.3 | 0.3 | 0.3 | 0.3 |
| Residual Dropout | 0.1 | 0.1 | 0.1 | 0.1 | 0.1 |
| MLP Dropout | 0.1 | 0.1 | 0.1 | 0.1 | 0.1 |
| Optimizer | AdamW | AdamW | AdamW | AdamW | AdamW |
| Betas | [0.9, 0.95] | [0.9, 0.95] | [0.9, 0.95] | [0.9, 0.95] | [0.9, 0.95] |
| Learning Rate | 1e-4 | 1e-4 | 1e-4 | 1e-4 | 1e-4 |
| Transformer Weight Decay | 0.05 | 0.05 | 0.05 | 0.05 | 0.1 |
| Other weight decay | 0.05 | 0.05 | 0.05 | 0.05 | 0.1 |
| Batch Size | 512 | 512 | 512 | 512 | 512 |
| Train Steps in Thousands | 30 | 25 | 15 | 30 | 300 |
| $\sigma_{max}$ | 80 | 80 | 80 | 80 | 80 |
| $\sigma_{min}$ | 0.001 | 0.001 | 0.001 | 0.001 | 0.001 |
| $\sigma_t$ | 0.5 | 0.5 | 0.5 | 0.5 | 0.5 |
| EMA | True | True | True | True | True |
| Time steps | Exponential | Exponential | Exponential | Exponential | Exponential |
| Sampler | DDIM | DDIM | DDIM | DDIM | DDIM |
| Parameter Count (Millions) | 460 | 460 | 460 | 460 | 685 |

Table 3: Summary of all the Hyperparameters for the MoDE policy used in our experiments.

### A.1    PRETRAINING DETAILS

| Dataset | Weight |
|---|---|
| BC-Z | 0.258768 |
| LIBERO-10 | 0.043649 |
| BRIDGE | 0.188043 |
| CMU Play-Fusion | 0.101486 |
| Google Fractal | 0.162878 |
| DOBB-E | 0.245176 |
| **Total** | 1.000000 |

Table 4: Dataset sampling weights used for training MoDE on a small subset of trajectories. The total dataset consists of 196k trajectories.

We pretrain a large variant of MoDE on a subset of available datasets from the Open-X-Embodiment Collaboration et al. (2023) to study the generalization ability of MoDE. An Overview of the used dataset is summarized in Table 4. Our pretraining dataset comprises 196K trajectories from 6 different sources, with weighted sampling across BC-Z (0.259), LIBERO-10 (0.044), BRIDGE (0.188), CMU Play-Fusion (0.101), Google Fractal (0.163), and DOBB-E (0.245). This dataset includes demonstrations from diverse robot platforms including Google robots, Franka Pandas, and Hello-Stretch robots, covering a wide range of manipulation tasks. The pretraining was conducted on 6 NVIDIA A6000 GPUs with 40GB VRAM each over 3 days, completing 300K training steps. We used a batch size of 1024 with a learning rate of 1e-4 and weight decay of 0.1. To ensure balanced dataset mixing during training, we implemented a large shuffle buffer of 400K samples. Each dataset was individually normalized to account for different scales and ranges across the various robot platforms. This diverse pretraining significantly improved MoDE's zero-shot generalization, particularly on challenging benchmarks like CALVIN ABC→D where we achieved a new state-of-the-art performance of 4.01 average rollout length. For reproducibility, we will release our pretrained model weights and preprocessing code to the community.

| Benchmark | MoDE | DP-T | DP-CNN | Avg. Baseline | Improvement |
|---|---|---|---|---|---|
| CALVIN ABC→D (norm.) | **0.678** | 0.226 | 0.270 | 0.248 | +151.1% |
| CALVIN ABCD→D (norm.) | **0.860** | 0.396 | 0.632 | 0.514 | +36.1% |
| LIBERO-90 | **0.910** | 0.690 | 0.780 | 0.735 | +16.7% |
| LIBERO-10 | **0.920** | 0.510 | 0.730 | 0.620 | +26.0% |
| Average Improvement Over Second-Best: | | | | | **57.5%** |

Table 5: Detailed Performance Improvement Analysis. CALVIN scores are normalized by dividing by 5 to align with LIBERO scale. Improvement calculated as: (MoDE - Avg. Baseline) / Avg. Baseline × 100. Final average is the mean of improvements across all four benchmarks compared to the second best Diffusion Policy variant on each one.

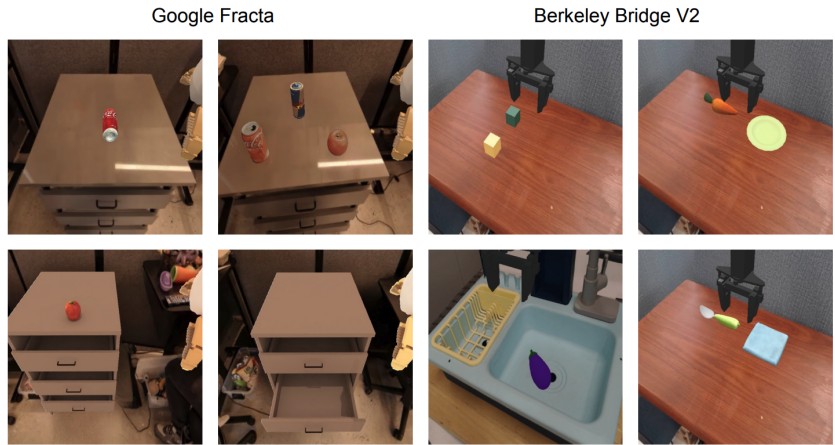

Figure 7: Example Scenes of the SIMPLER Li et al. (2024b) benchmark used to test generalist policies on various tasks from the Bridge and Google Fractal dataset.

For finetuning we freeze the routers of the model and remove the load-balancing loss and train on the local domain for 10k steps on LIBERO and 15k steps for CALVIN with a batch size of 64 per GPU and 4 GPUs.

## A.2 EXPERIMENTS DETAILS

**Average Performance Increase.** To quantify MoDE's advantages over existing Diffusion Policies, we compared its performance against the second-best method across all benchmarks. MoDE demonstrated substantial improvements, particularly in challenging transfer scenarios like CALVIN ABC→D where it outperformed the next best method by 151.1%. Even on the more standardized LIBERO benchmarks, MoDE maintained significant advantages of 16.7% to 26.0%. Averaging across all tasks, MoDE achieved a 57.5% improvement over the second-best performing method while maintaining its computational efficiency with 90% fewer FLOPs compared to dense tranformers with similar number of parameters.

### A.2.1 MoDE EVALUATION ON SIMPLER

We evaluate MoDE's capabilities as a generalist policy by comparing it against two state-of-the-art models trained on substantially larger datasets from Open-X-Embodiment: Octo (800K trajectories) (Octo Model Team et al., 2023) and OpenVLA (Kim et al., 2024)(1M trajectories). We conduct this comparison using the SIMPLER benchmark, which provides real2sim variants of the BridgeV2 (Walke et al., 2023) and Google Fractal datasets used to train RT-1 (Brohan et al., 2023). The benchmark encompasses diverse manipulation tasks across multiple environments, as illustrated in Figure 7.

| Metric | OpenVLA | | Octo Base | | MoDe (ours) | |
|---|---|---|---|---|---|---|
| | Score | Rank | Score | Rank | Score | Rank |
| Drawer Open | **16%** | **1** | 0% | 3 | 4.23% | 2 |
| Drawer Close | 20% | 2 | 2% | 3 | **34.92%** | **1** |
| Pick Can Horizontal | **71%** | **1** | 0% | 3 | 33.78% | 2 |
| Pick Can Vertical | 27% | 2 | 0% | 3 | **29.78%** | **1** |
| Pick Can Standing | **65%** | **1** | 0% | 3 | 36.44% | 2 |
| Move Near | **48%** | **1** | 3% | 3 | 30% | 2 |
| Drawer Open | 19% | 2 | 1% | 3 | **21.30%** | **1** |
| Drawer Close | 52% | 2 | 44% | 3 | **76.85%** | **1** |
| Pick Can Horizontal | **27%** | **1** | 21% | 3 | 22% | 2 |
| Pick Can Vertical | 3% | 3 | 21% | 2 | **40%** | **1** |
| Pick Can Standing | 19% | 2 | 9% | 3 | **35%** | **1** |
| Move Near | **46%** | **1** | 4% | 3 | 45.42% | 2 |
| Partial Put Spoon on Tablecloth | 4% | 3 | **35%** | **1** | 29.17% | 2 |
| Put Spoon on Tablecloth | 0% | 3 | 12% | 1 | **12.5%** | **1** |
| Partial Put Carrot on Plate | 33% | 2 | **53%** | **1** | 29.17% | 3 |
| Put Carrot on Plate | 0% | 3 | 8% | 1 | **8.33%** | 1 |
| Partial Stack Green Block on Yellow Block | 12% | 2 | **32%** | **1** | 8.33% | 3 |
| Stack Green Block on Yellow Block | 0% | 2 | 0% | 2 | 0% | 2 |
| Partial Put Eggplant in Basket | 8% | 3 | **67%** | **1** | 37.5% | 2 |
| Put Eggplant in Basket | 4% | 3 | **43%** | **1** | 8.33% | 2 |
| **Average** | 23.70% | 1.95 | 17.75% | 2.1 | **26.30%** | **1.65** |

Table 6: Detailed comparison of MoDE against two sota Generalist Policies OpenVLA (Kim et al., 2024) and Octo (Octo Model Team et al., 2023) tested on all SIMPLER tasks with 2952 evals.

The results of the evaluations are summarized in Table 6. On average MoDE achieves the highest average success rate of 26.30% and the best average ranking of 1.65 across all tasks, surpassing both Octo (17.75% success, 2.1 rank) and the 7.7B parameter OpenVLA model (23.70% success, 1.95 rank). MoDE shows particularly strong performance on challenging manipulation tasks like drawer manipulation (34.92% on drawer close) and precise object interactions (40% on vertical can picking). While specialized tasks like stacking blocks remain challenging for all models, MoDE's consistent performance across diverse tasks demonstrates its effectiveness as a scalable architecture for generalist policies.

### A.2.2 CALVIN BENCHMARK

The CALVIN benchmark (Mees et al., 2022b) is an established IL benchmark for learning language-conditioned behavior from human play data. In contrast to other benchmarks the data does not contain structured demonstrations, where the robot completes one task, but instead, the dataset was collected by humans, that randomly interact with the environment. From these long-horizon trajectories across the 4 different settings, the authors randomly cut out short sequences with 64 frames and labeled them with the task label. While the dataset offers models to train on the unlabeled part too, we restricted MoDE to only train on the labeled parts. The Franke Emika Panda robot is controlled using a Delta-End-Effector Space with a discrete gripper. We use two cameras to encode the current scene: a static camera and a wrist one and predict the next 10 actions, before receiving the next observations and generating another set of 10 actions.

**CALVIN ABC.** We train MoDE and our dense transformer baseline for 25k training steps with a batch size of 512 on a 4 GPU Cluster Node with 4 A6000 NVIDIA GPUs for 6.5 hours with all 1000 rollouts at the end of training. We report the mean results averaged over 3 seeds as done in all relevant prior work. All baselines are reported from the original paper given the standardized evaluation protocol of CALVIN (Mees et al., 2022b).

**CALVIN ABCD.** We train MoDE and our dense transformer baseline for 30k training steps with a batch size of 512 on a 4 GPU Cluster Node with 4 A6000 NVIDIA GPUs for 7.5 hours with all 1000 rollouts at the end of training. We report the mean results averaged over 3 seeds as done in all relevant prior work.

| Model | Block Push | Relay Kitchen | CAL ABC | CAL ABCD | L-10 | Average |
|-------|-----------|---------------|---------|----------|------|---------|
| Dense T | 0.96±0.02 | 3.73±0.12 | **2.83±0.19** | 4.13±0.11 | 0.91±0.02 | 0.839±0.144 |
| Token-Router | 0.97±0.01 | **3.85±0.03** | 2.67±0.04 | 4.29±0.08 | 0.90±0.01 | 0.845±0.161 |
| $\sigma_t$-Router | *0.97±0.01* | 3.79±0.04 | *2.79±0.16* | **4.30±0.02** | **0.92±0.02** | **0.851±0.151** |

Table 7: Overview of the performance of all different token routing strategies used for MoDE across 5 benchmarks. We mark the best result for each environment in **bold** and the second best in *cursive*. We use CAL to represent CALVIN. To average the results, we normalize all scores and compute the average over all environments.

### A.2.3 LIBERO BENCHMARK

**LIBERO-10.** The LIBERO-10 benchmark consists of 50 demonstrations for 10 different tasks that are all labeled with a text instruction. The Franka Emika Panda robot is controlled using an end-effector controller. Similar to CALVIN all models have access to two camera inputs: a static one and a wrist camera. We train MoDE and our dense transformer baseline for 50 epochs with a batch size of 512 on a 4 GPU Cluster Node with 4 A6000 NVIDIA GPUs for 2 hours with all 200 rollouts at the end of training. The benchmark does require to test models on 10 different long-horizon tasks. We test each task 20 times for each model and report the final average performance overall 10 tasks.

**LIBERO-90.** The LIBERO-10 benchmark consists of 50 demonstrations for 90 different tasks that are all labeled with a text instruction. The Franka Emika Panda robot is controlled using an end-effector controller. We train MoDE and our dense transformer baseline for $50k$ steps with a batch size of 512 on a 4 GPU Cluster Node with 4 A6000 NVIDIA GPUs for 12 hours with all 1800 rollouts at the end of training. The benchmark does require to test models on 90 different tasks in many different environments. We test each task 20 times for each model and report the final average performance overall 90 tasks.

### A.3 BASELINES

Below we explain several baselines used in the experiments in detail:

**Diffusion Policy-CNN/T** Inspired by (Chi et al., 2023), we evaluate extension of the DDPM based Diffusion Policy framework for goal-conditioned Multi-task learning. We evaluate two versions: the CNN-based variant and the Diffusion-Transformer variant, that is conditioned on context and noise using cross-attention. For our experiments we also use EDM-based Diffusion framework for fair comparison against MoDE. We optimized the ideal number of layers and latent dimension for the Transformer baseline and our final version uses 8 layers with a latent dimension of 1024. Larger or smaller variants resulted in lower average performance.

**RoboFlamingo.** RoboFlamingo (Li et al., 2024a) is a Vision-Language-Models (VLM) finetuned for behavior generation. The authors use a 3 billion parameter Flamingo model (Alayrac et al., 2022) and fine-tune it on CALVIN by freezing the forward blocks and only fine-tuning a new Perceiver Resampler module to extract features from a frozen vision-transformer image encoder and the cross-attention layers to process the image features. Finally, a new action head is learned to generate actions. Overall, the finetuning requires training approx. 1 billion of the parameters. We report the reported results from the paper since they use the standard CALVIN evaluation suite.

**SuSIE.** This model first finetunes Instruct2Pix, an image-generation diffusion model, that generates images conditioned on another image and a text description (Brooks et al., 2023) on the local CALVIN robotics domain and uses it as a high-level goal generator. The low-level policy is a CNN-based Diffusion Policy, that predicts the next 4 actions given the current state embedding and desired sub-goal from the high-level policy (Black et al., 2023).

**GR-1** A causal GPT-Transformer model (Wu et al., 2024), that has been pretrained on large-scale generative video prediction of human videos. Later, the model is finetuned using co-training of action prediction and video prediction on CALVIN. We report the results directly from their paper for the CALVIN benchmark.

## A.4 RECTIFIED FLOW FINETUNING FOR MoDE

We investigate the compatibility of MoDE with alternative denoising objectives by combining our architecture with Rectified Flow (RF) (Albergo & Vanden-Eijnden, 2022; Lipman et al., 2022). Using our OXE-pretrained MoDE model finetuned with an RF objective, we evaluate performance with reduced inference steps on the ABC→D setting.

| Method | Steps | No. Instructions in a Row (1000 chains) | | | | | |
| --- | --- | --- | --- | --- | --- | --- | --- |
| | | 1 | 2 | 3 | 4 | 5 | Avg. Len. |
| MoDE | 10 | 96.2% | 88.9% | 81.1% | 71.8% | 63.5% | 4.01±0.04 |
| MoDE-RF | 2 | 93.5% | 86.2% | 77.5% | 68.8% | 60.5% | 3.87±0.06 |

Table 8: Performance comparison of MoDE with standard vs RF objective on CALVIN ABC→D. MoDE-RF achieves comparable performance with just 2 denoising steps.

As shown in Table 8, MoDE-RF maintains strong performance while requiring only 2 denoising steps compared to the original 10 steps. The modest performance drop (4.01 vs 3.87 average rollout length) suggests that our noise-specialized experts successfully adapt to different denoising objectives. Future work could explore combining MoDE with Rectified Flow to further reduce inference steps while maintaining high performance.

## A.5 AVERAGE FLOPs COMPUTATION AND INFERENCE SPEED

We provide an in-depth comparison of the total parameters and FLOPs used for every method in Table 9. Additionally, we provide the computational efficiency metric per GFLOPS ($10^9$ FLOPS) to compare the various methods and measure the average prediction time for a single action. In the following, we detail the average GFLOPS computation for all relevant baselines on the CALVIN ABC benchmark. Specifically, we compare the average GFLOPS required to predict a single action.

To guarantee a fair comparison we evaluated all methods on the same NVIDIA A6000 GPU with 40 GB VRAM. To compute the average inference speed, we tested each method 100 times and removed large outliers to compute an average time.

**MoDE.** We benchmark the large pretrained variant with 12 layers, 4 experts and a hidden dim of 1024. The average GFLOPS for a forward pass are 0.7 GFLOPS. Without router caching, the model would require 5 GFLOPS, indicating that the router caching reduces the overall computational cost by over 90%. The architecture processes 14 tokens in total (1 noise + 1 goal + 2 images + 10 noisy action actions). MoDE predicts a sequence of 10 actions with 10 denoising passes. For the variant with ResNet-50, the image encoder requires 8.27 GFLOPS . On average for a single action, MoDE requires 10 forward passes with the transformer and a single pass with the ResNet-50. Consequently, the pretrained variant of MoDE needs 1.53 GFLOPS on average to predict a single action. The inference time for that model depends on the hardware. We measure an average inference time per action of 12.2.

**DP-CNN/T.** DP-CNN utilizes 0.8 GFLOPS for an average forward pass. The ResNet-18s require 3.62 GFLOPS. The model predicts 10 actions with 10 denoising steps and executes 10 actions without replanning. This results in the CNN version requiring 1.28 GFLOPS to predict a single action. For the Transformer version, the architecture predicts 10 actions using 10 denoising steps and processes 14 tokens in total (1 noise + 1 goal + 2 images + 10 noisy action actions) similar to MoDE. It achieves an average GFLOPS usage of 1.8 GFLOPS per forward pass through the transformer. The DP-T baseline requires 2.16 GFLOPS on average to predict a single action. The CNN version requires 11.7 ms to predict a single action on average and the transformer variant with its cross-attention conditioning needs 16.2 ms.

**RoboFlamingo.** For computational analysis, the model requires 34 GFLOPS to encode a single image with a ViT. For the policy backbone, we evaluated the "mpt-1b-redpajama-200b-dolly" variant as used in the paper. This architecture requires 656 GFLOPS for an average sequence of 32 tokens per forward pass. While multiple variants of RoboFlamingo exist, this provides a rough estimate of the required GFLOPS. In total, we estimate an average of 690 GFLOPS to predict an action in CALVIN. To predict a single action, the model requires 65 ms on average.

| Method | Active Params (M) | Total Params (M) | GFLOPS | PrT | Avg. Length | SF-Ratio | Inf. Time [ms] |
|---|---|---|---|---|---|---|---|
| Diff-P-CNN | 321 | 321 | 1.28 | × | 1.35 | 1.05 | **11.7** |
| Diff-P-T | 194 | 194 | 2.16 | × | 1.13 | 0.53 | 16.2 |
| RoboFlamingo | 1000 | 1000 | 690 | ✓ | 2.47 | 0.004 | 65 |
| SuSIE | 860+ | 860+ | 60 | ✓ | 2.69 | 0.045 | 199 |
| GR-1 | **130** | **130** | 27.5 | ✓ | 3.06 | 0.11 | 12.6 |
| **MoDE (ours)** | 436 | 740 | 1.53 | ✓ | **4.01** | **2.6** | 12.2 |

Table 9: Comparison of total and active number of parameters of methods used in the CALVIN benchmark. Additional overview of average FLOPS required by the different methods together with their average performance on the ABC benchmark. SF-Ratio compares average rollout length with GFLOPS.

**SuSIE.** In our computational analysis, we tested Instruct2Pix with 50 denoising steps as implemented by SuSIE. The resulting 1026 GFLOPS are divided by 20 as the model only generates new subgoals every 20 timesteps. The low-level policy uses a ResNet-50 image encoder with 8.27 GFLOPS. Contrast to other policies, SuSIe only computes one image per state and predicts actions every timestep. These are then averaged using exponential averaging. Thus, we omit the small diffusion head to get an estimate of 60 GFLOPS per action. For the average inference speed we meassure the time to generate a single goal image and divide it by 20, next we add the average time to encode two images with the ResNet-50 and 10 forward passes through a small MLP. Every 20 timesteps, when SuSIE generates a new image it requires 3777.62 ms for a single action generation. Otherwise its a lot faster with 10.7 ms. On average SuSIE requires 199 ms to generate a single action, which make it the slowest policy overall.

**GR-1.** The pretrained MAE Vision Transformer requires approx. 17.5 GFLOPS to encode a single image. The transformer backbone processes 150 tokens with a history length of 10 and 15 tokens per timestep (10 image tokens + 1 goal token + 1 proprioceptive token + 2 video readout tokens + 1 action token per timestep). Consequently, the average GFLOPS for a single action prediction utilizing the decoder with a latent dimension of 384 and 12 layers are 10 GFLOPS. In total, this results in an average computational cost of 27.5 for predicting a single action in CALVIN. For the average single action prediction the model requires 12.6 ms.

**Analysis.** Overall, MoDE offers the best performance to GFLOPS ratio across all tested baselines. Although MoDE is significantly larger in total size compared to the other Diffusion Policy architectures it requires similar inference speed and a low FLOP count. Additionally, it has demonstrated superior efficiency in terms of computational resources while maintaining high performance on the CALVIN benchmark tasks. For inference speed MoDE is the second fastest although it has a high total parameter count.

## A.6 DETAILED EXPERIMENTAL RESULTS

We summarize the ablations regarding the choice of routing in Table 7. Therefore, we test two 2 different routing strategies across 5 benchmarks.

## A.7 STATE-BASED EXPERIMENTS

We conduct additional experiments with MoDE on two established multi-task state-based environments:

**Relay Kitchen.** We utilize the Franka Kitchen environment from (Lynch et al., 2019) to evaluate models. This virtual kitchen environment allows human participants to manipulate seven objects using a VR interface: a kettle, a microwave, a sliding door, a hinged door, a light switch, and two burners. The resulting dataset consists of 566 demonstrations collected by the original researchers, where each participant performed four predetermined manipulation tasks per episode. The Franka Emika Panda robot is controlled via a 9-dimensional action space representing the robot's joint and end-effector positions. The 30-dimensional observation space contains information about the current state of the relevant objects in the environment. As a desired goal state, we randomly sample future states as a desired goal to reach.

For this experiment, we train all models for 40k training steps with a batch size of 1024 and evaluate them 100 times as done in prior work (Shafiullah et al., 2022; Cui et al., 2023; Reuss et al., 2023) to guarantee a fair evaluation. All reported results are averaged over 4 seeds. We train our models on a local PC RTX with an RTX 3070 GPU for approx. 2 hours for each run with the additional experimental rollouts.

**Block Push.** The PyBullet environment features an XArm robot tasked with pushing two blocks into two square targets within a plane. The desired order of pushing the blocks and the specific block-target combinations are sampled from the set of 1000 demonstrations as a desired goal state. The demonstrations used for training our models were collected using a hard-coded controller that selects a block to push first and independently chooses a target for that block. After pushing the first block to a target, the controller pushes the second block to the remaining target. This approach results in four possible modes of behavior, with additional stochasticity arising from the various ways of pushing a block into a target. The models only get a credit, if the blocks have been pushed in the correct target position and order. We consider a block successfully pushed if its center is within 0.05 units of a target square.

All models were trained on a dataset of 1000 controller-generated demonstrations under these randomized conditions. All models have been trained for 60k steps with a batch size of 1024. To evaluate them we follow prior work (Shafiullah et al., 2022; Cui et al., 2023; Reuss et al., 2023) and test them on 100 different instructions and report the average result over 4 seeds. We train our models on a local PC RTX 3070 GPU for approx. 3 hours for each run with a final evaluation. Demonstrations are sourced from a scripted oracle, which first pushes a randomly chosen block to a selected square, followed by the other block to a different square. The policies are conditioned to push the blocks in the desired configuration using a goal state-vector. We chose an action sequence length of 1 given a history length of 4 for these experiments, which are inspired by our dense diffusion transformer baseline BESO (Reuss et al., 2023).

**Baselines.** In this setting, we compare MoDE against several SOTA goal-conditioned policies. We test two transformer architectures, C-BeT (Cui et al., 2023) and VQ-BeT (Lee et al., 2024a), that predict discretized actions with an offset. C-BeT uses k-means clustering together with an offset vector while VQ-BeT leverages residual Vector Quantization to embed actions into a hierarchical latent space. Further, we test against a dense diffusion policy transformer model BESO (Reuss et al., 2023). BESO uses the same continuous-time diffusion policy combined with a dense transformer to predict a single action given a sequence of prior states. To enable a fair comparison, we chose the same hyperparameters for BESO and MoDE in both settings. We test all models averaged over 4 seeds and report the mean values directly from prior work (Lee et al., 2024a).

|  | Block Push | Relay Kitchen |
|---|---|---|
| C-BeT | 0.87±(0.07) | 3.09±(0.12) |
| VQ-BeT | 0.87±(0.02) | 3.78±(0.04) |
| BESO | 0.96±(0.02) | 3.73±(0.05) |
| **MoDE** | **0.97±(0.01)** | **3.79±(0.02)** |

Table 10: Comparison of the performance of different policies on the state-based goal-conditioned relay-kitchen and block-push environment averaged over 4 seeds. MoDE outperforms the dense transformer variant BESO and other policy representations on all baselines.

**Results.** The results of both experiments are summarized in Table 10. MoDE achieves a new SOTA performance on both benchmarks and outperforms the dense transformer variant of BESO in both settings. Further, MoDE achieves higher performance compared to other policy representation methods such as VQ-BeT and C-BeT.

### A.7.1 MIXTURE-OF-EXPERTS ABLATIONS

**Q: How does the Load Balancing Loss influence the Expert Distribution?**

We analyze how the load balancing loss affects expert distribution across noise levels by training MoDE on LIBERO-10 with varying load balancing weights $\gamma_{LB} \in [0.1, 0.01, 0.001, 0.0001]$ Figure 9 visualizes the resulting expert distributions.

With a high load balancing loss of 0.1, experts are used almost uniformly across all layers with minor variations in two out of eight layers (Figure 9a). However, this enforced uniformity comes at a cost -

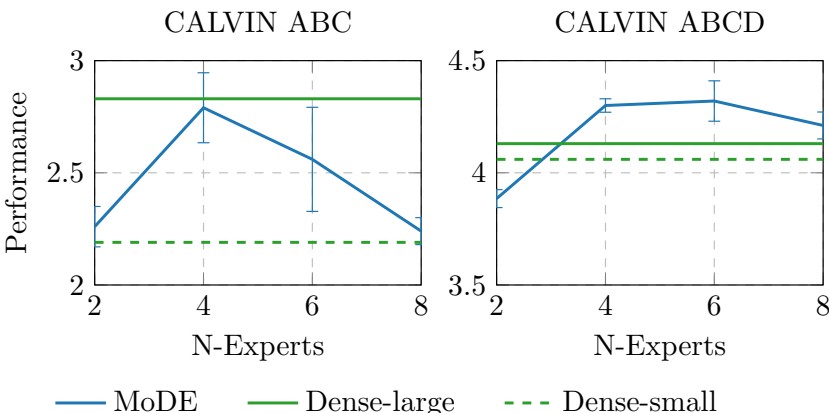

Figure 8: Scaling performance of MoDE and Dense-MoDE on CALVIN ABC and ABCD environments, showing average performance for 2 to 8 experts using best-performing variants for each environment.

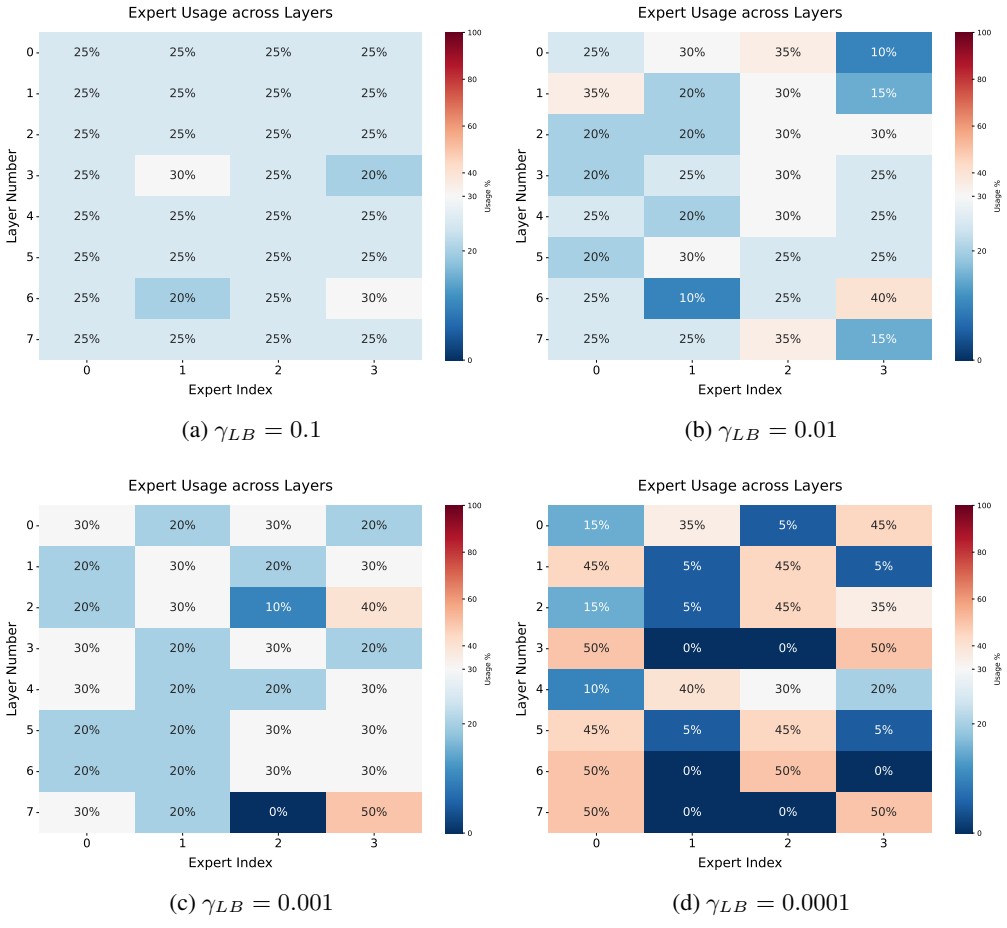

Figure 9: Average Expert Utilization for different Load Balancing Weights across all denoising levels.

the average performance drops to 0.9. This result suggests that enforcing equal expert usage across noise levels may constrain the model's learning capacity.

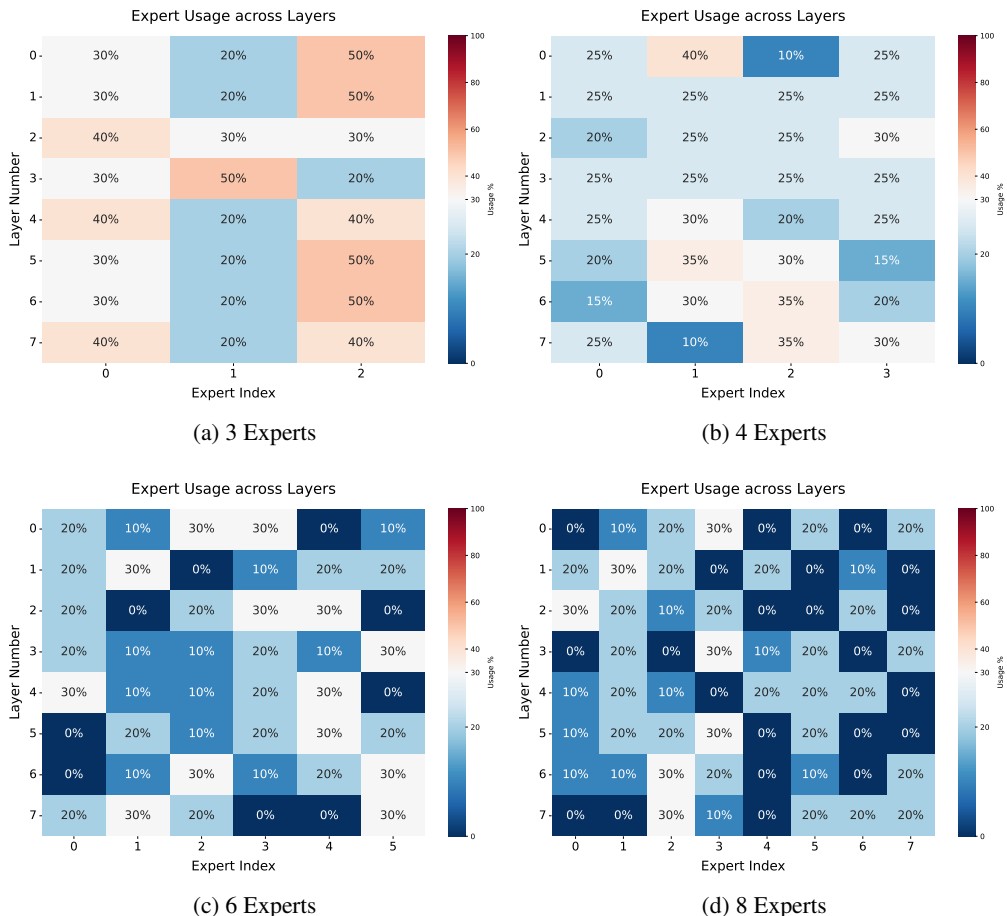

Figure 10: Average Expert Utilization for different number of used experts across all denoising levels.

At $0.01$, we observe a more flexible distribution while maintaining good overall expert utilization (Figure 9b). Within individual layers, expert usage varies from $10\%$ to $40\%$ in various layers. This enables for specialization while preventing any expert from becoming dominant. This configuration achieves the best average performance and provides evidence that moderate specialization benefits the policy most.

As we reduce $\gamma_{LB}$ further to a range of $[0.001, 0.0001]$, the expert distribution gets worse and the model there is an expert collapse is happening for lb of 0.0001 in several layers (Figure 9c and Figure 9d). These values show, that a the load balancing weight is crucial to guarantee a good usage of all experts for the model. The lower performance of these models (0.86 for $\gamma_{LB} = 0.001$ and 0.83 for $\gamma_{LB} = 0.0001$) suggests that balanced expert participation is important for optimal Diffusion Policy performance.

**Q: What happens with increased number of Experts?** Next, we study the load balancing distribution for models that use more experts. Our experiments with 6 and 8 expert variants on LIBERO-10 reveal challenges in scaling beyond 4 experts. As shown in Figure 10, configurations with more experts consistently underutilize their capacity, with most layers actively using only 4 experts even with increased load balancing regularization ($\gamma = 0.1$). These larger configurations also show decreased performance, indicating that 4 experts provides an optimal balance between representation capacity and learning efficiency. We hypothesize that additional experts impair the model's ability to learn effective representations from image and language inputs while maintaining noise-level specialization. This phenomenon persists even when increasing the load balancing loss factor, further supporting our finding that 4 experts offers the best trade-off between model capacity and performance.

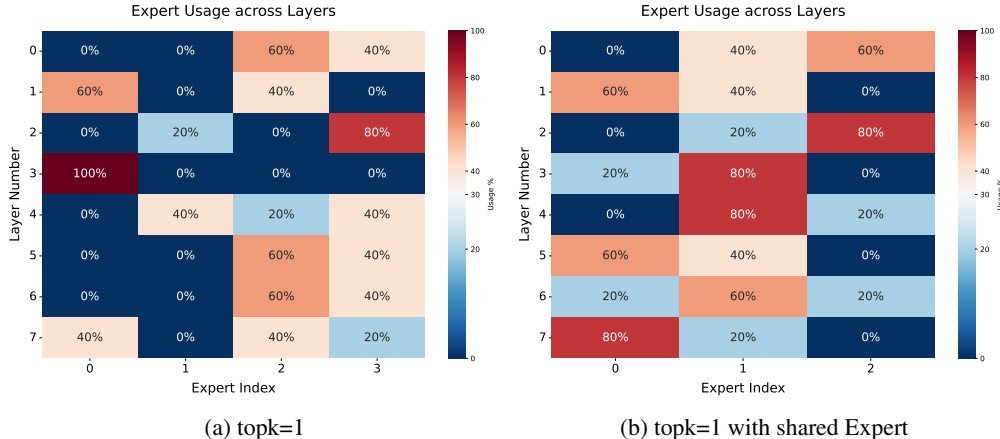

(a) topk=1        (b) topk=1 with shared Expert

Figure 11: Average Expert Utilization for two variants that use topk=1: left MoDE with 4 experts, right MoDE with 4 experts, where 1 is used in all settings as a shared one. Overall, MoDE struggles to equally distribute tokens across experts in this setting.

**Q: How is the expert distribution with topk=1?** We further study the impact of using $top_k = 1$ for MoDE. We analyze MoDE's behavior with topk=1 routing, both with and without a shared expert, using 4 experts total. As visualized in Figure 11, topk=1 configurations struggle to maintain balanced expert utilization, typically collapsing to using only two of the four available experts. While this configuration shows only modest performance degradation on LIBERO-10, its impact is more visible on the challenging CALVIN ABC benchmark, where the average rollout length drops from 3.34 (topk=2) to 2.72 (topk=1). These results highlight the importance of expert interpolation across noise levels, with topk=2 providing the necessary flexibility to blend expert specializations for optimal performance.

**Q: What expert distribution patterns emerge for different noise levels?**

Finally, we analyze the different experts loads for various noise levels of our biggest model after pretraining on diverse robotics data. Detailed results are shown in Figure 12.

Our analysis of the expert distribution reveals several key insights about how the model learns to organize its mixture-of-experts architecture:

- **Noise-Level Specialization**
    - Experts specialize between high-noise ($\sigma_1$-$\sigma_7$) and low-noise ($\sigma_8$-$\sigma_{10}$) regimes, particularly visible in L4 where E3 and E0 show distinct high activation patterns in different noise regimes
    - Clear transition point around $\sigma_8$ where expert utilization patterns shift, most evident in L4 and L5 with stark changes in expert activation
    - Smooth handoff between denoising phases, exemplified in L7 where expert activations gradually transition across noise levels

- **Layer-wise Organization**
    - Early layers (L0-L4) show distinct expert specialization, particularly visible in L4's strong alternating expert patterns
    - Middle layers (L5-L7) demonstrate more distributed expert utilization, shown by more balanced activation patterns
    - Later layers (L8-L11) return to clearer specialization, evident in L11's distinct expert preferences
    - First layer (L0) shows particularly strong specialization for low-noise scenarios, with clear expert preferences in the final denoising steps

- **Expert Role Distribution**
    - Different experts develop specialized roles, most clearly visible in L4 where experts show strong preferences for specific noise ranges

- Some experts consistently handle high-noise denoising (evident in L3 and L4) while others focus on low-noise refinement (for example in in L7-L8)

- "Transitional experts" emerge in middle noise ranges, particularly visible in L5's balanced activation patterns

- Model employs different expert combinations across layers, well demonstrated in the contrasting patterns between L4 and L8

- **Load Balancing Properties**

    - Effective load balancing achieved across experts, particularly visible in the distributed patterns of L5-L7

    - No single expert dominates across all noise levels, as shown by the varied activation patterns in each layer

    - Smooth transitions between noise regimes, most evident in the gradual activation changes in L4 and L5

These findings provide strong evidence that MoDE effectively partitions the denoising process across its experts, with each expert naturally specializing in different phases. The emergence of this organization without explicit supervision supports the effectiveness of our architectural choices and routing strategy.

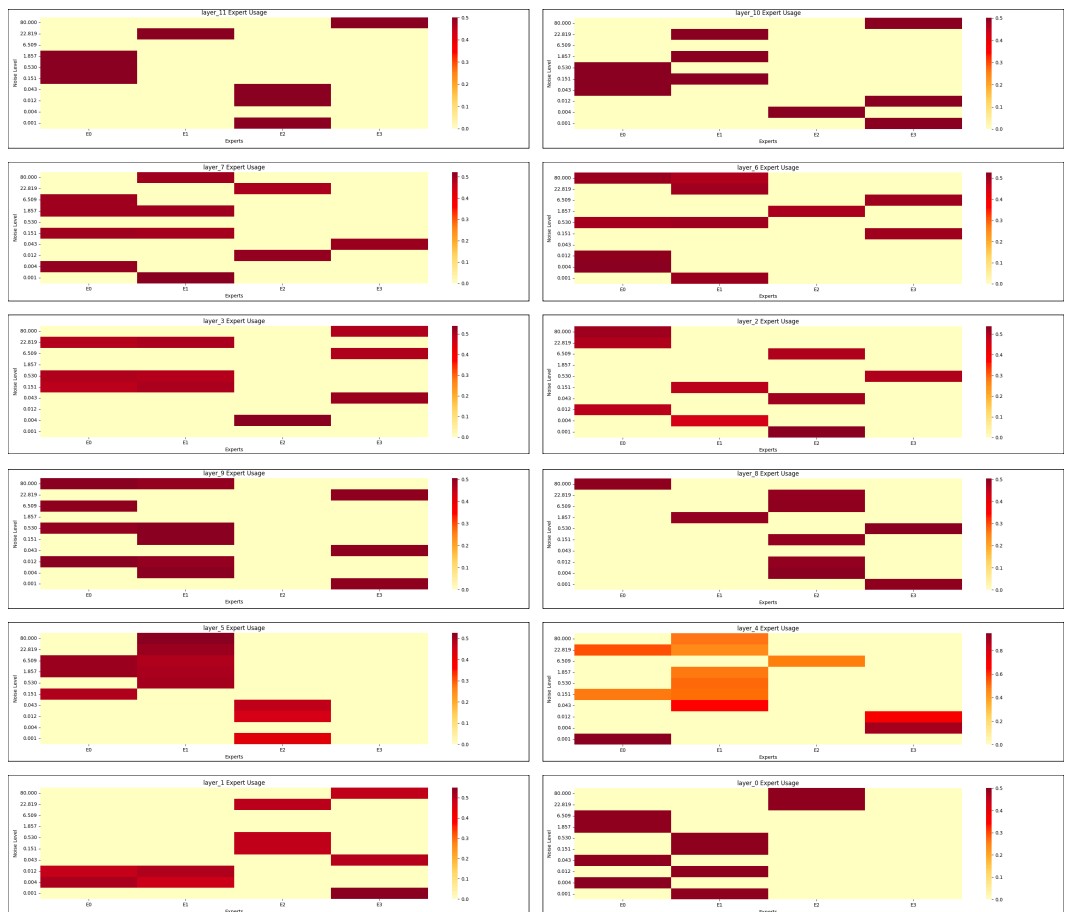

Figure 12: This image shows the expert usage distribution across 12 layers in a Mixture of Experts (MoE) model. Each subplot represents a different layer (from layer 0 to layer 11), with experts labeled E0, E1, E2, and E3 on the x-axis. The y-axis indicates the log-scaled token count, displaying the frequency of tokens routed to each expert within that layer. The color gradient indicates the proportion of tokens assigned to each expert, where darker colors represent higher usage.

## A.8 EXTENDED RELATED WORK

**Mixtre-of-Experts in Robotics.** In the context of robotics, MoE models are used in many settings without being combined with a transformer architecture. Several works use a mixture of small MLP policies, that focus on different skills in Reinforcement Learning (Obando-Ceron et al., 2024; Celik et al., 2022; 2024) or for robot motion generation(Hansel et al., 2023; Le et al., 2023), another body of work utilizes combinations of small CNNs robot perception (Riquelme et al., 2021; Mees et al., 2016). Further applications include learning multimodal behavior using a mixture of Gaussian policies (Blessing et al., 2023; Li et al., 2023a). Despite the extensive usage of MoE in many domains, no prior work has tried to utilize MoE together with Diffusion Policies for scalable and more efficient Diffusion Policies.

**Transformers for Robot Learning.** Transformer models have become the standard network architecture for many end-to-end robot learning policies in the last few years. They have been combined with different policy representations in the context of IL. One area of research focuses on generating sequences of actions with Variational Autoencoder (VAE) models (Bharadhwaj et al., 2023; Zhao et al., 2023). These action-chunking transformer models typically use an encoder-decoder transformer as a policy architecture. Several Diffusion Policies, such as Octo (Octo Model Team et al., 2023), BESO (Reuss et al., 2023), ChainedDiffuser (Xian et al., 2023) and 3D-Diffusion-Actor leverage a transformer model as a policy backbone. Another direction of research treats behavior generation as discrete next-token predictions similar to auto-regressive language generation (Touvron et al., 2023). C-Bet, RT-1, and RT-2 use discretized action binning to divide seen actions into $k$-classes (Cui et al., 2023; Shafiullah et al., 2022; Brohan et al., 2022; Zitkovich et al., 2023), while VQ-BeT (Lee et al., 2024a) learns latent actions with residual Vector Quantization. Several works have shown the advantages of using pre-trained LLM or VLM as a policy backbone, which are then finetuned for action generation (Brohan et al., 2023; Gu et al., 2024; Collaboration et al., 2023; Li et al., 2024a). None of the recent work considers using any Mixture-of-Expert architecture for policy learning. MoDE is the first architecture to leverage MoE architecture combined with diffusion for behavior generation.

