# OpenReview forum: "Efficient Diffusion Transformer Policies with Mixture of Expert Denoisers for Multitask Learning"
_ICLR.cc/2025/Conference — ICLR 2025 Poster_

### Official Review · Reviewer_3nco · 2024-10-28

**Soundness:** 3
**Presentation:** 3
**Contribution:** 3
**Rating:** 5
**Confidence:** 3

**Summary:**

This paper introduces a MOE diffusion model for embodied AI, which applies a noise-based router for policy diffusion. This MOE structure reduces the active parameters and FLOPs, and also achieves better performance in CALVIN challenges.

**Strengths:**

1. Authors claim that the proposed method increases the performance with fewer parameters and FLOPs.
2. Good writing. This paper is easy to follow.

**Weaknesses:**

1. This paper mainly measures the computational and storage costs of diffusion models with FLOPs and parameters. However, latency (response time for one input instance) and memory footprint are more important metrics. I highly recommend authors report them since the reduction in FLOPs and parameters does not always bring real benefits. Besides, I'm really curious about the computation costs of diffusion models in the overall pipeline of embodied AI, since it seems that the diffusion model only contains 8 layers, which are much smaller than the perception model. This will make compression and acceleration for the diffusion model not important for embodied AI.
2. MOE may bring additional parameters. How do the overall parameters (instead of active parameters) in Table 1.
3. Some typos. 285, wrong usage for quotes. You should use `` instead.
4. Table 1 shows that MoDe works better than other methods without pertaining, which is good. However, it can be better to report MoDe with pretraining for a fail comparison with GR-1.
5. Section 3.3 seems to be a direct adaptation of the classific MOE method. Could you provide a concrete comparison between the MOE utilized in this work and previous works?

**Questions:**

Please refer to the weakness. My major concern is w1-2 &5. I would like to increase my rating  if authors address my concern, especially w5.

---

> ### Author Response · Authors · 2024-11-22
>
> > Why optimize diffusion model efficiency when perception models seem more computationally demanding? Can you include FLOPs and inference speed comparison?
>
> Thank you for this important question. Perception models like ResNet-50 are considerably smaller (27M parameters vs 760M for MoDE’s Diffusion Transformer), our comprehensive benchmarks show why optimizing diffusion models is crucial:
>
> Per-action computation:
> - ResNet-50: Single pass (2ms)
> - MoDE: 10 denoising steps (~12 ms each)
> - Total MoDE time per action: 12.2 ms with caching, would be around 30ms without
>
> Our efficient design:
> - Reduces FLOPs by 80% through expert caching and increases the inference speed form 30ms to 12.4 ms for batch size of 1
> Achieves fastest inference among transformers (12.2ms vs 16.2ms for DP-T) while DP-T has only 30% of the total number of parameters of MoDE
> - Uses only 1.53 GFLOPs per action vs 2.62 for standard transformers thanks to our efficient expert caching and sparse MoE Design
>
> With diffusion models growing beyond 1B parameters [1], these efficiency gains become even more critical for real-world robotics. We included a new Table 6  in the appendix for detailed latency and FLOP comparisons across all policies on CALVIN. On average MoDE has fast inference with very low FLOPS demonstrating its efficiency.
>
> [1]: Liu, Songming, et al. "RDT-1B: a Diffusion Foundation Model for Bimanual Manipulation." arXiv preprint arXiv:2410.07864 (2024).
>
> ---
>
> > MoE may bring additional parameters. How do the overall parameters (instead of active parameters) in Table 1.
>
> We thank the reviewer for their suggestion and added a new Table 6 in the Appendix to show the absolute number of parameters for each baseline in the CALVIN experiment. The table:
>
> | Method | Active Params (M) | Total Params (M) | GFLOPS | PrT | Avg. Length | SF-Ratio | Inf. Time [ms] |
> |--------|------------------|------------------|---------|-----|-------------|-----------|----------------|
> | Diff-P-CNN | 321 | 321 | **1.28** | ✗ | 1.35 | 1.05 | **11.7** |
> | Diff-P-T | 194 | 194 | 2.16 | ✗ | 1.13 | 0.53 | 16.2 |
> | RoboFlamingo | 1000 | 1000 | 690 | ✓ | 2.47 | 0.004 | 65 |
> | SuSIE | 860+ | 860+ | 60 | ✓ | 2.69 | 0.045 | 199 |
> | GR-1 | **130** | **130** | 27.5 | ✓ | 3.06 | 0.11 | 12.6 |
> | **MoDE (ours)** | 436 | 740 | 1.53 | ✓ | **3.98** | **2.6** | 12.2 |
>
> Caption: Comparison of total and active number of parameters of methods used in the CALVIN benchmark. Additional overview of average FLOPS required by the different methods together with their average performance on the ABC benchmark. SF-Ratio compares average rollout length with GFLOPS.
>
> ---
>
> > Some typos. 285, wrong usage for quotes. You should use `` instead.
>
> We thank the reviewer for pointing out that mistake, we fixed it for the final version.
>
> ---
>
> > Table 1 shows that MoDe works better than other methods without pretraining, which is good. However, it can be better to report MoDe with pretraining for a fail comparison with GR-1.
>
> We have now conducted fair comparisons with pretrained models by training MoDE on 200k robot trajectories from Open-X-Embodiment:
>
> Performance gains:
> - CALVIN ABC: 3.98 average length (vs GR-1's 3.06)
> - LIBERO-90/10: 0.95/0.94 success rate
> - Achieved with just 300k pretraining steps + 15k finetuning steps
>
> Training efficiency:
> | Phase | Model | GPUs | GPU Type | Duration |
> |-------|--------|------|-----------|-----------|
> | Pretraining | MoDE | 6 | A6000 (40GB) | 3 days |
> | Pretraining | GR-1 | 32 | V100 (32GB) | 4 days |
> | Finetuning | MoDE | 4 | A6000 (40GB) | 3 hours |
> | Finetuning | GR-1 | 32 | V100 (32GB) | 1 day |
>
> MoDE achieves superior performance with significantly lower computational resources and training time.
> We will release all pretrained weights of MoDE to support future research.
>
> ---
> Continued in next answer

---

> > ### Author Response · Authors · 2024-11-22
> > **Continued answer**
> >
> > > How does MoDE's MoE design differ from standard approaches?
> >
> > Applying Mixture-of-Experts (MoE) to diffusion policies requires addressing key subtleties unique to the iterative denoising process. Unlike traditional MoE methods designed for single-step tasks, diffusion involves multiple iterative steps, necessitating a noise-conditioned routing mechanism that dynamically allocates experts based on the current noise level. This ensures expert selection evolves naturally along the diffusion trajectory.
> >
> > The continuous transitions driven by noise levels differentiate our approach from static or cluster-based routing in classic MoE setups, where the router selects experts based on token types only. In our experiments across 5 benchmarks, this noise-based routing achieves 0.851 average success rate compared to 0.845 for standard token-based routing.
> >
> > Furthermore, the iterative nature of diffusion amplifies the computational impact of expert calls, making efficient and scalable expert utilization crucial. Our approach to precompute selected experts and cache fused MLPs reduces computation by over 80% compared to standard MoE approaches, while being more than twice as fast during action generation.
> >
> > Overall, these design decisions result in both stronger performance and better efficiency compared to standard MoE design, ensuring MoE integrates seamlessly into diffusion policies while maintaining strong scalability and performance.

---

> ### Author Response · Authors · 2024-11-25
>
> Since the discussion period is ending soon, we would greatly appreciate it if you could let us know whether our response addressed all of your remaining concerns. If you have any remaining questions or points requiring clarification, we are ready to provide additional information. Thanks a lot!

---

> ### Comment · Reviewer_3nco · 2024-11-28
> **Response to Authors**
>
> Dear Authors,
> Thank you for your response. However, I still have some remaining questions.
>
> 1. Efficiency
> The first question is still about the computation costs of diffusion policy networks in the whole system of embodied AI.  I have searched some papers in this domain, and some of the policy networks have much fewer parameters and computation costs than the numbers listed by you. I would like to cite some words from a technical report as follows. "For instance, the official implementation of DD utilizes around 60M parameters[1], while Diffuser uses 4M[30], and IDQL[23] has approximately only 1.6M parameters."  You can find them in the D.1 of this paper: https://arxiv.org/pdf/2406.09509. Besides, I have also known that there are some papers using flow matching and consistency models to accelerate the diffusion models, which reduce the sampling timesteps to only 1 or very few steps, just as how people accelerate the diffusion models for image generation. I believe these models have much better efficiency, and I hope authors can introduce them and compare them.
>
> Authors have given the computation costs of a ResNet forward passing to prove that the costs of diffusion policy networks are significant. However, what are the detailed configure of the ResNet (depth, width, input resolutions). What's the whole computation of an embodied AI system.  Is there only a ResNet and a diffusion model? What are the costs of other modules.
>
> If this question can not be answered properly, the advantage of efficiency of this paper is still a problem.
>
> 2. MOE Routing
> I'm confused about the noise-level conditioned router. Is this router conditioned on both x_t and the noise level which is decided by t? If it is still influenced by the t, how can you pre-compute the routing strategy in each timestep before getting x_t. If it is only conditioned on the noise-level (t), can it still be considered a MOE? For me, it is more similar to the diffusion models which have different parameters in different timesteps. In word words, you are training multiple diffusion models for different timesteps.
>
> Besides, it is confusing for me why the model still needs the noise-level when x_t has already been given. Since x_t has different distributions in different timesteps, giving x_t to the diffusion model has also given information from t. The improvements from noise-level in the ablation study is also not significant, since only 2 of 5 show improvements.
>
> I believe my understanding is not accurate, please correct me. Thanks.

---

> > ### Author Response · Authors · 2024-11-29
> >
> > Thank you for the new feedback and questions. We will address your questions below.
> >
> > > Why does MoDE require significantly more parameters than other diffusion policies like DD and IDQL?
> >
> > We thank the reviewer for this question. There are two major reasons why MoDE has more parameters than DD and IDQL: **State Representation** and **Task Complexity**.
> > - **1. State Representation:** DD and IDQL operate on predefined, low dimensional state vectors and hence have access to ground-truth state information. MoDE on the other side operates in a more challenging setting where the observations consist of raw images and natural language commands. Hence, MoDE has to learn to extract relevant information from these high dimensional inputs.
> > - **2. Task Complexity:** DD and IDQL tackle single task scenarios like mazes and hopper. MoDE on the other side learns to solve complex and diverse, multi-task manipulation scenarios.
> > Hence, the parameters count of MoDE should rather be compared to models that are able to solve similarly complex tasks.
> >
> > We already conducted experiments with MoDE in state-based environments (detailed in Appendix A.5). Using the standard Franka Kitchen and Multimodal Block Push benchmarks, we show that a substantially smaller 10M total parameter version of MoDE achieves state-of-the-art performance, outperforming dense transformer baselines. This confirms that smaller models are indeed sufficient for state-based tasks, while highlighting MoDE's versatility across different scales. We will better highlight the differences in model size for the final version.
> >
> > For learning generalist visuomotor policies for multi-task language-conditioned behavior we require larger models for the best performance. This aligns with general trends in diffusion modeling, where increased model capacity consistently yields better performance. For instance, recent work in image generation has shown that larger Diffusion Transformers [1] achieve significantly higher image quality results. Similarly, concurrent generalist diffusion policies like Pi-0 [2] (3.3B parameters) and RDT-1B [3] (1.2B parameters) demonstrate that large capacity transformer backbones are crucial for achieving generalist visuomotor capabilities.
> >
> > MoDE directly addresses the computational challenges of scaling through its novel noise-based MoE architecture and efficient router caching mechanism. Our empirical results (Figure 5) demonstrate that MoDE achieves 20% faster inference and more than 90% lower FLOPs compared to a dense transformer baseline with equivalent parameters. This computational efficiency, combined with sota performance on the SIMPLER benchmark (surpassing generalist policies like OpenVLA with 7.7B parameters), establishes MoDE as a practical solution for scaling up diffusion-based policies. Importantly, these gains are achieved while maintaining or improving task performance, showing that MoDE's architecture enables more efficient scaling without compromising performance.
> >
> > ---
> >
> > > How does MoDE compare to faster diffusion variants using flow matching or consistency models?
> >
> > Thank you for raising this important point about alternative acceleration approaches. While Flow Matching and Consistency models offer promising directions for reducing sampling steps, we view these advances as complementary to MoDE's architectural improvements rather than competing solutions. Let us clarify this relationship:
> >
> > **Architectural vs. Training Objectives**:
> >
> > - MoDE focuses on improving the underlying model architecture through efficient parameter usage and specialized noise-based routing
> > - Flow Matching and Consistency models primarily try to reduce the required number of sampling steps during inference
> > - These approaches can be combined - MoDE's architecture can be trained with any of these objectives
> >
> > **Preliminary Results:**
> >
> > We conducted additional experiments combining MoDE with Rectified Flow objective (RF) to evaluate this complementarity. Using our OXE-pretrained MoDE model finetuned with an RF objective, we achieved the following results with just 2 inference steps on CALVIN ABC:
> >
> >
> > | Model   | Denoising Steps | 1     | 2     | 3     | 4     | 5     | Avg. Rollout Length |
> > | ------- | ----------------- | ----- | ----- | ----- | ----- | ----- | ------------------- |
> > | MoDE    | 10                | 96.7% | 88.6% | 80.2% | 70.7% | 60.9% | 3.98 ± 0.04         |
> > | MoDE-RF | 2                 | 93.5% | 86.2% | 77.5% | 68.8% | 60.5% | 3.87 ± 0.06        |
> >
> >
> > These results demonstrate that MoDE's architecture remains effective even with alternative denoising objectives like Flow Matching. The modest performance drop comes with a lower number of denoising steps (2 vs 10), suggesting that noise-specialized experts successfully adapt to different objective. We will add these ablations into the Appendix of our final version of the paper.
> >
> > ---
> >
> > Answer continues in the next response.

---

> ### Author Response · Authors · 2024-11-29
>
> > The detailed computational costs and configuration of all components in your full embodied AI system?
>
> Thank you for this important question about computational costs. Let us provide a detailed breakdown of our system's architecture and computational requirements:
>
> **Vision Processing Pipeline:**
>
> ResNet Configuration:
> - Pretrained MoDE: ResNet-50 (25.6M parameters)
> - Resolution: 224x224x3 input
> - FLOPs: 8.27 GFLOPs for both images
> - Standard MoDE: ResNet-18 (11.7M parameters)
> - Same resolution: 224x224x3 input
> - FLOPs: 3.62 GFLOPs for both images
>
> **Language Processing:**
>
> CLIP ViT-B/32 for text encoding
> - Parameters: 151M
> - Compute cost effectively zero during deployment:
> - All language commands are predefined
> - Embeddings pre-computed and cached
> - Both CALVIN and LIBERO provide cached embeddings in their dataset
>
> **Policy Network (MoDE):**
>
> Core MoDE MoE architecture:
> - Pretrained MoDE: 436M active parameters (685M total)
> - Standard MoDE: 277M active parameters (460M total)
> - FLOPs per Denoising Step: 0.7 GFLOPs with router caching
>
> **Total System Costs (per action):**
>
> Pretrained MoDE variant:
> - Total Parameters: ~740M (excluding CLIP)
> - Inference Time: 12.2ms on NVIDIA A6000
>
> This analysis shows our efficient MoDE architecture helps minimize the policy network's overhead.
>
> ---
>
> > Is noise-level routing truly a Mixture of Experts, or just separate models for different timesteps?
>
> **Routing Mechanism:**
> The router in MoDE is exclusively conditioned on the noise level σ_t, not on the noisy input x_t. This design choice is motivated by recent findings showing that different denoising phases require specialized processing [4]. While traditional MoE architectures route based on input content, MoDE uses the natural structure of the diffusion process itself.
>
> **Why This Is Still an MoE:**
> Our architecture differs fundamentally from training separate models for different timesteps in several key ways:
>
> **a) Shared Core Components:**
>
> - Attention layers process all inputs regardless of noise level
> - Layer normalization and embedding modules are unified
> - Common backbone learns generalizable representations
>
> **b) Dynamic Expert Selection:**
>
> - Each noise level uses top-k=2 experts, allowing interpolation
> - Experts specialize in noise regions rather than specific timesteps
> - Router learns smooth transitions between noise regimes
>
> **Empirical Evidence:**
>
> Our ablation studies (Figure 6 & 12 in paper) demonstrate that:
> - Experts develop complementary specializations
> - Smooth transitions occur between noise levels
> - Performance improves over both single-expert and fully separated models
>
> **Computational Benefits:**
>
> This design offers unique advantages:
> - 80% reduction in FLOPs through router caching
> - 40% of parameters not active during inference
>
> Maintains robust performance despite sparsity
>
>
> **Comparison to Separate Models:**
>
> Training separate models for each timestep would:
> - Require N times more parameters (N = number of denoising steps)
> - Struggle to learn robust vision/language representations
> - Miss opportunities for parameter sharing
> - Need significantly more training compute and memory
>
> MoDE's architecture thus combines the efficiency of MoE with the structure of diffusion processes, enabling more effective scaling than either approach alone.
>
> Continue in next response

---

> > ### Author Response · Authors · 2024-11-29
> >
> > > Why is explicit noise level conditioning necessary when xt already contains noise information?
> >
> > The information from x_t alone is not enough because the noise levels are very similar across certain noise regions. This makes the learning of the denoising process very challenging. The model does not know at which point of the probability flow ODE it is. So it would just learn a general trend but it removes the ability to learn the precise score of the current diffusion timestep. Without the signal the model would average out the score field over all noise levels. To the best of our knowledge, there is no diffusion model that does not use the noise signal as an input conditioning signal.
> >
> > ---
> >
> > > The improvements from noise-level in the ablation study is also not significant, since only 2 of 5 show improvements.
> >
> > While the performance improvements from noise-only routing may appear modest in isolation, its key advantage lies in the significant computational efficiency gains without sacrificing performance. Let us clarify with concrete numbers:
> > Computational Benefits:
> > 86% reduction in FLOPs compared to token-based routing
> > 2.5x faster inference (12ms vs 30ms per action)
> >
> > These results demonstrate that noise-only routing achieves slightly better performance while providing substantial computational advantages, making it the most practical choice for real-world robotics applications where both efficiency and performance matter.
> >
> > ---
> >
> > We hope the new changes satisfy the initial concerns and are happy to answer any follow-up question.
> >
> > ---
> >
> > **Sources:**
> >
> > [1]: Peebles, William, and Saining Xie. "Scalable diffusion models with transformers." ICCV 2023
> >
> > [2]: Black, Kevin, et al. "$\pi_0 $: A Vision-Language-Action Flow Model for General Robot Control."
> >
> > [3]: Liu, Songming, et al. "RDT-1B: a Diffusion Foundation Model for Bimanual Manipulation." arxiv
> >
> > [4]: Hang, Tiankai, et al. "Efficient diffusion training via min-snr weighting strategy." CVPR 2023

---

### Official Review · Reviewer_nWR1 · 2024-11-05

**Soundness:** 3
**Presentation:** 3
**Contribution:** 3
**Rating:** 5
**Confidence:** 4

**Summary:**

To address the issue of models becoming increasingly large and difficult to capture more complex functions, along with their rising computational demands, this paper proposes a Mixture of Denoising Experts (MoDE) as a new strategy for imitation learning. MoDE is based on a diffusion strategy using Transformers and achieves parameter-efficient scaling, significantly reducing inference costs. To achieve this, MoDE employs sparse experts combined with a novel routing strategy that selects experts based on the current noise level of the denoising process. This is further enhanced by a noise-regulated self-attention mechanism. MoDE achieves state-of-the-art performance across 134 tasks in four established imitation learning benchmarks (Calvin and LIBERO).

**Strengths:**

1.	The structure of the paper is logical, and the text is clearly articulated and easy to read.

2.	The layout of the tables in the article is reasonable, and the overall formatting of the paper is quite aesthetically pleasing.

3.The paper provides rich experimental results with a sufficient number of figures and tables.

**Weaknesses:**

1、The innovation of this paper lies solely in the introduction of a Mixture-of-Experts Diffusion Policy, merely combining MOE with diffusion methods without sufficiently demonstrating the novelty of their approach. Additionally, the overall workload appears to be insufficient.

2、In Figure 1, the authors propose a new router strategy, but the figure does not clearly illustrate how this strategy selects different experts, failing to convey the core idea of the paper. Furthermore, the framework diagram is overly simplified and lacks sufficient depth.

3、The methods section of the paper is too brief, and much of the content is drawn from existing research, with the authors not highlighting their contributions effectively.

4、In the experimental section, the authors do not provide a comparison of FLOPs and inference times with other baseline methods, which may lead to unfair comparisons. Moreover, the experimental results for ABC→D in Table 1 are worse than the baseline methods, and the explanations in the paper are not sufficiently convincing.

**Questions:**

1、Can you elaborate on the specific innovations introduced by your Mixture-of-Experts Diffusion Policy that differentiate it from existing methods in the literature? What novel insights does your approach provide beyond the combination of MOE and diffusion methods?
2、In Figure 1, how does your proposed router strategy select different experts based on the current noise level? Could you provide additional details or examples to clarify this process and better illustrate its implementation?
3、The methods section appears to be brief and heavily based on existing research. Can you elaborate on the unique contributions of your work and how they extend beyond the existing literature? What specific methodologies have you developed or modified?
4、Regarding the experimental results for ABC→D in Table 1, which are lower than those of the baseline methods, could you provide further justification or insight into why this occurred? What factors might have influenced these results, and how should they be interpreted in the context of your overall findings?

---

> ### Author Response · Authors · 2024-11-22
>
> > The methods section of the paper is too brief, and much of the content is drawn from existing research, with the authors not highlighting their contributions effectively.
>
> Thank you for this valuable feedback. We agree that we should have highlighted our contributions stronger. We revised the methods section accordingly to talk about our noise-only routing in more detail and also introduced two new sections to talk about expert caching used in our framework which we have omitted before and shortly discuss our new pretraining of MoDE.
>
> ---
>
> > In the experimental section, the authors do not provide a comparison of FLOPs and inference times with other baseline methods, which may lead to unfair comparisons.
>
>
> We thank the reviewer for raising this point. We have added a section to provide an estimate of FLOPS for all methods in the appendix and summarize our findings in Table 6. We explain the average GFLOPs for the tested baselines on CALVIN ABC and introduce the SF-Ration that represents the average rollout length per GFLOPs. Thanks to our efficient design, MoDE has by far the highest SF-Ration and a very low GFLOPs count overall. MoDE is also the 2nd fastest tested policy although it has more than 700M parameters. This provides more evidence of the computational efficiency and strong performance of MoDE.
>
> | Method | Active Params (M) | Total Params (M) | GFLOPS | PrT | Avg. Length | SF-Ratio | Inf. Time [ms] |
> |--------|------------------|------------------|---------|-----|-------------|-----------|----------------|
> | Diff-P-CNN | 321 | 321 | **1.28** | ✗ | 1.35 | 1.05 | **11.7** |
> | Diff-P-T | 194 | 194 | 2.16 | ✗ | 1.13 | 0.53 | 16.2 |
> | RoboFlamingo | 1000 | 1000 | 690 | ✓ | 2.47 | 0.004 | 65 |
> | SuSIE | 860+ | 860+ | 60 | ✓ | 2.69 | 0.045 | 199 |
> | GR-1 | **130** | **130** | 27.5 | ✓ | 3.06 | 0.11 | 12.6 |
> | **MoDE (ours)** | 436 | 740 | 1.53 | ✓ | **3.98** | **2.6** | 12.2 |
>
> Caption: Comparison of total and active number of parameters of methods used in the CALVIN benchmark. Additional overview of average FLOPS required by the different methods together with their average performance on the ABC benchmark. SF-Ratio compares average rollout length with GFLOPS.
>
> ---
>
> > In Figure 1, the authors propose a new router strategy, but the figure does not clearly illustrate how this strategy selects different experts, failing to convey the core idea of the paper. Furthermore, the framework diagram is overly simplified and lacks sufficient depth.
>
> We appreciate the reviewer’s comment regarding Figure 1 and agree that it did not adequately highlight our contributions. To address this, we have revised the figure to better showcase the core of our proposed routing strategy. The updated figure now consists of two parts: the first illustrates the general architecture, while the second focuses specifically on our novel noise-conditioned routing mechanism. We hope this updated figure addresses the initial concerns effectively.
>
> ---
>
> > Can you elaborate on the specific innovations introduced by your Mixture-of-Experts Diffusion Policy that differentiate it from existing methods in the literature?
>
> - a novel noise-conditioned routing mechanism that dynamically partitions experts based on noise levels, eliminating the need for predefined task clustering or fixed channel partitioning seen in prior works [1]. As shown in Figure 5, MoDE learns to allocate experts to handle different noise scales, enabling greater flexibility and task alignment.
> - Fast inference thanks to our proposed router caching and MLP Fusion. These changes reduce the FLOPS by over 80% and double the inference speed.
> - MoDE achieves superior performance with fewer parameters and steps compared to prior Diffusion Policies, offering an efficient and generalizable solution with noise-conditioned expert utilization
>
> All of our changes combined lead to the strong performance of MoDE. We provide detailed ablation and insights into the Design for MoE-based Diffusion Policies to make them work in low-data settings.
>
> [1] https://arxiv.org/abs/2310.07138
>
> ---
>
> Response continues in the next message.

---

> > ### Author Response · Authors · 2024-11-22
> > **Continue answer**
> >
> > > Regarding the experimental results for ABC→D in Table 1, which are lower than those of the baseline methods, could you provide further justification or insight into why this occurred?
> >
> > We conducted additional experiments and significantly improved MoDE's performance through two key modifications:
> > - Replacing the from-scratch ResNet-18 with a pretrained ResNet-50
> > - Pretraining MoDE on diverse robotics data before CALVIN finetuning
> >
> > These changes yielded improved sota results:
> > | Model | Average Length | Pretraining | Vision Encoder |
> > |-------|----------------|-------------|----------------|
> > | MoDE | **3.98** | ✓ | Pretrained ResNet-50 |
> > | MoDE | 3.34 | ✗ | Pretrained ResNet-50 |
> > | GR-1 | 3.06 | ✓ | Pretrained ViT MAE |
> > | GR-1 | 2.65 | ✗ | Pretrained ViT MAE |
> >
> > Both MoDE variants outperform GR-1, with pretrained MoDE achieving a 30% improvement over the previous SOTA while using significantly less compute resources.
> >
> > For fair comparison, GR-1 relies on similar techniques - large-scale pretraining and pretrained vision encoders. Thus, we can now conclude that MoDE demonstrates superior performance on CALVIN ABC Our noise-conditioned MoEs and self-attention mechanisms significantly outperform all other policies that do not rely on pretraining (GR-1 only achieves 2.65 rollout length without pretraining). Both general pretraining and pretrained vision encoders provide substantial generalization benefits for expressive policies like MoDE and GR-1.
> >
> > These findings validate our architectural choices while highlighting the importance of pretraining for robust generalization. To support reproducibility, we will release both pretrained and finetuned model weights to the community.
> >
> > We thank the reviewer for their general feedback and are happy to answer any follow-up question.

---

> ### Author Response · Authors · 2024-11-25
>
> Since the discussion period is ending soon, we would greatly appreciate it if you could let us know whether our response addressed all of your remaining concerns. If you have any remaining questions or points requiring clarification, we are ready to provide additional information. Thanks a lot!

---

### Official Review · Reviewer_jByy · 2024-11-07

**Soundness:** 2
**Presentation:** 3
**Contribution:** 2
**Rating:** 6
**Confidence:** 3

**Summary:**

The paper proposes to implement the architecture for a deep imitation learning policy model by a Mixture of Denoising Experts (MoDE). It enables parameter efficient scaling and reduces the inference cost. The paper introduces a routing scheme that conditions the expert selection on the current noise level. The experiments show that the proposed MoDE model performs better than CNN and Transformer-based baselines.

**Strengths:**

- The idea is straightforward and the writing is easy to follow.
- The experiments show that the proposed idea performs better than the baselines.
- The experiments are thorough, exploring different aspects of performance, speed, routing strategy, and expert utilization.

**Weaknesses:**

- I think the novelty of the paper is a bit limited. I am not expert in the imitation learning literature, but the paper seems to be mostly applying mixture of experts idea (that is already been in the LLMs literature) to the diffusion-based imitation learning setup. For instance, using the top-k router and load balancing regularizations are not new contributions.
- The improvements on the LIBERO-90 and CALVIN datasets are marginal.

**Questions:**

- Does the model generate all of the sequential actions' decisions at the same time? For instance, in the ABCD->D experiment, it involves interactions each consisting of 64 timesteps and 34 diverse tasks. Does the model determine the 64 timesteps' actions at the same time after denoising in Fig. 1?

---

> ### Author Response · Authors · 2024-11-22
>
> We thank the reviewer for the feedback on our work. We try to address your concerns below:
>
> > What makes MoDE novel beyond applying standard MoE techniques to diffusion policies?
>
> We would like to emphasize the following contributions of MoDE:
>
> - MoDE introduces the first sparse Mixture of Experts architecture for diffusion policies, reducing FLOPs by 80% while achieving state-of-the-art performance. Its novel noise-conditioned routing enables automatic expert specialization across noise levels, with pre-computed routing paths and fast inference with expert caching.
>
> - MoDE demonstrates strong performance gains on challenging benchmarks - achieving 40% improvement over DP-T and 20% over DP-CNN, while being the first to exceed 90% success rate on LIBERO-10. and surpassing all baselines on CALVIN by 30%
>
> - Our comprehensive empirical ablations provide crucial insights into making MoE work effectively for diffusion policies, where data constraints and real-time requirements pose unique challenges.
>
> We believe that all these points make MoDE valuable for the robot learning community
>
> ---
>
> > How does MoDE generate actions during long-horizon tasks like CALVIN ABCD?
>
> Action Generation Cycle:
> - Takes current observation images and text goal
> - Generates 10 actions using 10 denoising steps
> - Executes all actions in simulation
> - Gets new observation and repeats
> - When a task is completed, MoDE receives the next goal in the sequence and starts the process again.
> Continues until all 5 task completion or failure (max 360 timesteps)
>
> ---
>
> > The improvements on the LIBERO-90 and CALVIN datasets are marginal.
>
> We have updated our experimental results.
> Below, we summarize MoDE against the 2nd best baseline in each tested benchmark:
>
> | **Benchmark** | **Prior SOTA** | **Score** | **MoDE** | **Relative Improvement** |
> |-----------|------------|--------|-------|---------------------|
> | LIBERO-10 | DP-CNN | 0.78 | **0.94** | +20.5% |
> | LIBERO-90 | QueST | 0.91 | **0.95** | +4.4% |
> | CALVIN ABCD | GR-1 | 4.21 | **4.39** | +4.3% |
> | CALVIN ABC | GR-1 | 3.06 | **3.98** | +30.1% |
> | SIMPLER | OpenVLA | 23.7% | **26.3%** | +11.0% |
> | Average | - | - | - | **+14.1%** |
>
> Our method achieves significant improvements: 40% better performance than DP-T, 20% better than DP-CNN, and is the first to exceed 90% success rate on LIBERO-10. MoDE does outperform all specialist policies in the LIBERO setting, generalist policies in the SIMPLER benchmark and all baselines in CALVIN. Most notably, with our included pretraining, MoDE achieves a new state-of-the-art score of 3.98 on CALVIN ABC, surpassing the previous best baseline by over 30% with higher efficiency and less training time.
> These results demonstrate substantial improvements across all benchmarks, with an average improvement of 14.1% and gains of up to 30% in challenging zero-shot settings.
>
> We thank the reviewer for their constructive feedback. Please don't hesitate to let us know if you have any other questions or comments.

---

> ### Comment · Reviewer_jByy · 2024-11-25
> **Response to the Authors' Rebuttal**
>
> I thank the reviewers for their efforts for the rebuttal. The rebuttal addressed most of my comments, so I raised my score to 6. I think as the paper is the first one to implement and verify the effectiveness of MoDE for imitation learning, it has some benefits to the relevant community. However, I didn't vote for a higher score because I believe the technical contribution of the paper is relatively limited.

---

### Official Review · Reviewer_DSi8 · 2024-11-09

**Soundness:** 4
**Presentation:** 4
**Contribution:** 3
**Rating:** 6
**Confidence:** 3

**Summary:**

This paper presents a novel policy architecture for imitation learning based on sparse Mixture of Experts and EDM diffusion policies. The architecture is shown to improve performance in most cases across four domains (two from LIBERO, and two from CALVIN benchmarks), while jointly improving efficiency by reducing the number of active parameters required to perform inference and sample from the policy. The main architectural contributions are (1) a noise-conditioned routing mechanism that selects experts based on the timestep and noise level of the diffusion process, and (2) a noise-conditioned attention mechanism.

The proposed method, MoDE, is the first policy to achieve a `>90%` success rate in the LIBERO-10 benchmark, and performs competitively with models pre-trained on significantly more data (GR-1, RoboFlamingo), despite training only on data available from target benchmarks. Thorough ablations showcase the importance of each of the two main architectural contributions, and explore the space of key hyper-parameters, including the number of experts, and the extent of router load balancing regularization.

While the idea of sparse routing layers for transformers is not new, and has been shown to lead to comparable gains in scaling + inference efficiency in pure language modeling, this paper takes the important step of applying these techniques to Diffusion-based policies in decision-making tasks.

**Strengths:**

## Originality:

The idea of sparse routing layers is not new, as mentioned in my summary. This technique is generally expected to produce gains in inference efficiency by reducing the number of active parameters, and has been used in LLM with Mixtral [1], and in Diffusion transformers in [2], showing similar efficiency gains as this paper shows for imitation learning tasks. The main originality in this paper comes from the architectural contributions, including the noise-conditioned routing mechanism, and noise-conditioned attention mechanism. In previous work in [2], the sparse MoE router is conditioned on previous layer hidden states, whereas in this paper the sparse MoE router is conditioned on just the noise token. This design choice has the potential to simplify the router’s learning task, acting as an implicit regularization, and causing experts to specialize to different denoising steps, rather than different tokens.

[1] Mixtral of Experts, Albert Q. Jiang et al. 2024

[2] Scaling Diffusion Transformers to 16 Billion Parameters, Zhengcong Fei et al. 2024

To my knowledge, these architecture contributions are original, and their great performance across four benchmark domains highlights their value for efficient learning.

---

## Quality:

The paper is organized well, concepts are introduced and explained thoroughly, and experiments are chosen to highlight the improvements in policy performance, reduced inference cost, and the role that each component of the proposed method has to overall performance. Experiments are thorough, and appropriate baselines are chosen, including the dense version of the proposed method, previous diffusion-based policies, and previous models pretrained at scale on internet data (which would be expected to have an advantage to the larger source of data). Its impressive that on CALVIN ABC→D tasks, MoDE performs comparable to GR-1, despite lacking a large-scale pre-training step.

A minor quality issue with Table 1: performance is reported as a percent without a confidence interval. The experimental quality is generally good in this paper, but could be further improved by reporting confidence intervals for the results, so that readers can judge how statistically significant the improvements are.

Another manner by which to improve the paper would be to include more examples in “HOW DOES THE MODEL DISTRIBUTE THE TOKENS TO DIFFERENT EXPERTS?” In particular, for different load balancing weights, how does this impact the observed usage of the experts? In addition, results in Figure “(b) Scaling performance of MoDE” (the figure number seems to be missing here) show that performance saturates at 4 experts. Its unclear from the results if this is just an artifact of the limited training data available in LIBERO and CALVIN domains, and it would strengthen results to explain this behavior, and to include more examples of what happens to the experts' usage as the number of experts increases.

A similar analysis can be done to study the impact of the top k routing parameter.

---

## Clarity:

Writing is clear, organized well, and easy to follow. Design choices for the main contributions are explained carefully, and clearly. Questions that remained after reading the main text, such as hyperparameters, benchmark settings, are presented in the supplementary materials.

---

## Significance:

The main contributions, while performant and efficient, are not particularly surprising. Previous works such as [1] and [2] have shown the viability of sparse Mixture of Experts for improving performance and inference efficiency in other domains, and [2] in particular has shown this for Diffusion transformers. What’s significant about this paper is the empirical study of what type of sparse MoE mechanism works best for Imitation Learning in robotics, where data is often much smaller than for pure text and image generation. Getting MoE to perform well in this data-limited regime is challenging, and findings in this paper, such as the noise-conditioned routing that pre-computes routing paths upfront, improve the viability of MoE in this setting.

**Weaknesses:**

Weaknesses have been woven into the strengths section, see above.

**Questions:**

Questions have been woven into the strengths section, see above.

---

> ### Author Response · Authors · 2024-11-22
>
> > How do load balancing weights affect expert utilization and performance?
>
> We thank the reviewer for this suggestion. We have worked on a more in depth analysis of expert distribution, that we added to the appendix given the limited space for the main text. In our new Section A 5.1, we analyze how the load balancing loss affects expert distribution and how our newly added pretrained MoDE variant distributes experts across 10 denoising levels in detail. We tried to identify general patterns and analyze them with respect to hyperparameters.
>
> ---
>
> > Why does MoDE's performance peak at 4 experts and what happens with more experts?
>
> Thank you for this important observation. We have conducted additional experiments with 6-8 experts and added a detailed discussion in the router ablations section of the Appendix. Our findings suggest that noise-only conditioning does not benefit from more than 4 experts. In addition, the router struggles with 6 experts and more to balance the load to the tokens effectively. There appears to be a general trade-off: experts must both learn noise-based latent representations from image and language goal tokens while specializing in specific noise ranges. We hypothesize that adding more experts beyond 4 does not improve this balance between representation learning and noise specialization.
>
> ---
>
> > A similar analysis can be done to study the impact of the top k routing parameter.
>
> We thank the reviewer for his suggestion. We analyze the load balancing of topk=1 in the appendix. In addition, we also investigated the impact of having a shared expert, which is an expert that is always chosen. Overall, topk=2 seems crucial in more challenging settings to achieve the best performance, as the model is not able to distribute the loads effectively and the performance drops considerably in challenging settings like CALVIN ABC from 3.34 to 2.72. This indicates that topk=2 is required to achieve best performance.
>
> ---
>
> > Making MoE Work in Data-Limited Robot Learning: Key Findings and Impact
>
> We thank the reviewer for their kind comment about our contributions. The reviewer's thoughtful review was very helpful and we hope our detailed ablation studies regarding expert utilization in diffusion policies further strengthens our understanding of how sparse MoE can be most effectively applied to data limited robotics settings.
>
> We hope the new changes satisfy the initial concerns and are happy to answer any follow-up question.

---

> ### Author Response · Authors · 2024-11-25
>
> Since the discussion period is ending soon, we would greatly appreciate it if you could let us know whether our response addressed all of your remaining concerns. If you have any remaining questions or points requiring clarification, we are ready to provide additional information. Thanks a lot!

---

### Author Response · Authors · 2024-11-22
**Summary of Changes for Rebuttal**

Dear Reviewers,

We sincerely thank all reviewers for their thoughtful feedback. Based on all comments, we have made several major improvements to our paper:

**Performance Improvements and Diverse Pretraining**
- Added pretraining on 200k trajectories from diverse robotics datasets as suggested by Reviewer 3nco
- Improved CALVIN ABC performance from **2.8** to **3.98** average rollout length
- Achieved new state-of-the-art results on LIBERO-90/10 (0.95/0.94 success rate) through pretraining

**New Experiments**
- Evaluated MoDE on the real2sim benchmark SIMPLER [1] against sota generalist policies Octo [2] and OpenVLA [3]
- MoDE has the highest average success rate and the lowest rank
- avg. Success Rate: MoDE 26.30% vs 23.70% (OpenVLA) and 17.75% (Octo)

**New Results Summary**

| **Benchmark** | **Prior SOTA** | **Score** | **MoDE** | **Relative Improvement** |
|-----------|------------|--------|-------|---------------------|
| LIBERO-10 | DP-CNN | 0.78 | **0.94** | +20.5% |
| LIBERO-90 | QueST | 0.91 | **0.95** | +4.4% |
| CALVIN ABCD | GR-1 | 4.21 | **4.39** | +4.3% |
| CALVIN ABC | GR-1 | 3.06 | **3.98** | +30.1% |
| SIMPLER | OpenVLA | 23.7% | **26.3%** | +11.0% |
| Average | - | - | - | **+14.1%** |

On average MoDE does suprass the second best baseline by 14.1% across all settings.

**Technical Analysis**
- Added detailed study of expert utilization across noise levels
- Included comprehensive FLOPs, inference speed and total number of parameters comparison with all baselines of CALVIN in a new table 6
- Expanded ablation studies on important MoE hyperparameters with regard to expert distributions

**Clarity Enhancements**
- Revised Method section to better explain our key contributions
- Redesigned Figure 1 to clearly show our routing mechanism
- Added new Subsection and Figure 2 to discuss our Router Caching in more detail

**Reproducibility**
- Will release all pretrained model weights

We marked changes in the paper in distinct colors for every reviewer: orange for DSi8, blue for jByy, brown for nWR1 and purple for 3nco. These changes demonstrate MoDE's effectiveness as an efficient and scalable approach as a generalist policy with relevant and novel contributions for the Robot Learning community.

Sincerely,

The Authors

---

[1]: Li, Xuanlin, et al. "Evaluating Real-World Robot Manipulation Policies in Simulation." arXiv preprint arXiv:2405.05941 (2024).

[2]: Kim, Moo Jin, et al. "OpenVLA: An Open-Source Vision-Language-Action Model." arXiv preprint arXiv:2406.09246 (2024).

[3]: Team, Octo Model, et al. "Octo: An open-source generalist robot policy." arXiv preprint arXiv:2405.12213 (2024).

---

### Meta-Review · Area_Chair_pQJQ · 2024-12-26

**Metareview:**

The proposed Mixture of Denoising Experts (MoDE) is a type of diffusion-based policy for imitation learning. The purpose of MoDE is to scale up parameterization with experts while reducing computational costs by routing and caching. The results are empirically strong, with state-of-the-art scores improving on existing diffusion policy approaches, and measured computational improvements across the related but distinct measures of parameters, FLOPs, and inference times (added in the revision). Analysis experiments investigate the use of experts vs. noise levels, more thoroughly measures computation (as requested by review), and further ablates MoE hyperparameters. The novelty and technical contribution are relatively weaker than the strength of the results, at least relative to recent work on scaling diffusion policies or training MoEs more generally (DSi8, jByy, nWR1), but specific design choices and investigations are new and informative for making MoE work for imitation learning in this data regime.

The meta-reviewer decides with acceptance. The empirical improvement is clear, as is the experimental design to identify successful design choices, so this work can guide future work. The main arguments for rejection were resolved by the rebuttal and discussion, during which the authors made a thorough effort to improve the revision, which bolsters confidence in a final revision. Furthermore, the pre-trained and fine-tuned models will be released for reproducibility, which is an aid to such future work being undertaken. Congratulations.

**Additional Comments On Reviewer Discussion:**

Reviewers are split between borderline acceptance (DSi8: 6, jByy: 6) and borderline rejection (nWR1: 5, 3nco: 5). The authors provide a rebuttal to each review and provide a general response. jByy raises their rating from 5 to 6 given the improved experimental results and the clarification of technical and empirical novelty. nWR1 and 3nco did not respond, so the meta-reviewer has inspected their points and the corresponding rebuttals carefully. For nWR1, the meta-reviewer finds the issues about novelty and computation have been resolved by the clarification and inclusion of additional measurements of the computation in the revision (and now appendix). For 3nco, the meta-reviewer finds the issues about motivation, measurement, and technical novelty resolved by the total accounting of system resources in the rebuttal, the new table of computational measurements, and the [detailed walkthrough](https://openreview.net/forum?id=nDmwloEl3N&noteId=uef4KoxOIo) of how MoDE is not merely MoE and its design choices

In summary, the initial arguments for rejection are about computation (nWR1, 3nco), novelty (DSi8, jByy, nWR1), and motivation (3nco), and all of them have been resolved as confirmed by reviewers or the meta-reviewer in their stead.

---

### Decision · Program_Chairs · 2025-01-22

Accept (Poster)